# Modeling the interinfluence of fertilizer-induced NH₃ emission, nitrogen deposition, and aerosol radiative effects using modified CESM2

Ka Ming Fung[1,a], Maria Val Martin[2], Amos P. K. Tai[1,3]

[1] Graduate Division of Earth and Atmospheric Sciences, The Chinese University of Hong Kong, Sha Tin, Hong Kong
[2] Leverhulme Centre for Climate Change Mitigation, Department of Animal & Plant Sciences, University of Sheffield, Sheffield, UK
[3] Institute of Environment, Energy and Sustainability, and State Key Laboratory of Agrobiotechnology, The Chinese University of Hong Kong, Sha Tin, Hong Kong
[a] Now at: Department of Civil and Environmental Engineering, Massachusetts Institute of Technology, Cambridge, MA, USA

*Correspondence to*: Ka Ming Fung (kamingfung@mit.edu) & Amos P. K. Tai (amostai@cuhk.edu.hk)

**Abstract.** Global ammonia (NH₃) emission is expected to continue to rise due to intensified fertilization for growing food to satisfy the increasing demand worldwide. Previous studies focused mainly on estimating the land-to-atmosphere NH₃ injection but seldom addressed the other side of the bidirectional nitrogen exchange – deposition. Ignoring this significant input source of soil mineral nitrogen may lead to an underestimation of NH₃ emissions from natural sources. Here, we used an Earth system model to quantify NH₃-induced changes in atmospheric composition and the consequent impacts on the Earth's radiative budget and biosphere, as well as the impacts of deposition on NH₃ emissions from the land surface. We implemented a new scheme into the Community Land Model version 5 (CLM5) of the Community Earth System Model version 2 (CESM2) to estimate the volatilization of ammonium salt (NH₄⁺) associated with synthetical fertilizers into gaseous NH₃. We further parameterized the amount of emitted NH₃ captured in the plant canopy to derive a more accurate quantity of NH₃ that escapes to the atmosphere. Our modified CLM5 estimated that 11 Tg-N yr⁻¹ of global NH₃ emission is attributable to synthetic fertilizers. Interactively coupling terrestrial NH₃ emissions to atmospheric chemistry simulations by the Community Atmospheric Model version 4 with chemistry (CAM4-chem), we found that such emissions favor the formation and deposition of NH₄⁺ aerosol, which in turn disrupts the aerosol radiative effect and enhances soil NH₃ volatilization in regions downwind of fertilized croplands. Our fully-coupled simulations showed that global-total NH₃ emission is enhanced by nitrogen deposition by 2.4 Tg-N yr⁻¹, when compared to the baseline case with 2000-level fertilization but without deposition-induced enhancements. In synergy with observations and emission inventories, our work





provides a useful tool for stakeholders to evaluate the intertwined relations between agricultural trends, fertilize use, $NH_3$ emission, atmospheric aerosols, and climate, so as to derive optimal strategies for securing both food production and environmental sustainability.

## 1. Introduction

Global NH3 emission has risen from 59 to 65 Tg-N yr$^{-1}$ during 2000–2008, driven mainly by the increasing fertilizer use and manure handling in farms and animal operations (Sutton et al., 2013). After entering the atmosphere, $NH_3$ gas readily neutralizes sulfuric acid ($H_2SO_4$) and nitric acid ($HNO_3$), which are derived from the oxidation of sulfur dioxide ($SO_2$) and nitrogen oxides ($NO_x$), forming inorganic sulfate-nitrate-ammonium (SNA) aerosols (Behera and

Sharma, 2012). These ammonium ($NH_4^+$) salts can constitute 25–75% of inorganic fine particulate matter ($PM_{2.5}$, particles with an aerodynamic diameter <2.5 µm) (Ianniello et al., 2011; Snider et al., 2016), which causes not only haze and smog that lower visibility, but also respiratory and cardiovascular diseases that harm human health (Tie and Cao, 2009; Xing et al., 2016; Yang et al., 2019). In 2010 alone, an estimated 2.6 million premature deaths were

associated with $PM_{2.5}$ pollution (Wang et al., 2017). Without proper controls, unbridled use of fertilizer to boost food production for the fast-growing population can further enhance global agricultural $NH_3$ emissions by ~12% in 2050 compared to year-2010 level, posing an even greater health risk via $PM_{2.5}$ formation (Bodirsky et al., 2014). Our public health system may have to spend 20–290 billion USD more each year to compensate for the $NH_3$-derived

detrimental effects on air quality and health (Gu et al., 2012; Paulot and Jacob, 2014; Guthrie et al., 2018).

Excessive atmospheric $NH_3$ also threatens ecosystems. The highly soluble $NH_3$ gas and aerosol $NH_4^+$ (together known as $NH_y$) eventually return to the Earth's surface via dry and wet deposition, thus modifying the terrestrial nitrogen cycle. $NH_y$ deposited on canopy foliage can

be taken up and become readily available to promote photosynthesis (Wortman et al., 2012), but if highly concentrated it can also injure plant tissues and suppress biomass growth (Fangmeierfl et al., 1994; Krupa, 2003). Though $NH_y$ deposition can enrich soil nutrients, it also brings several adverse effects, including soil acidification and forest biodiversity loss (Tian and Niu, 2015; Lu et al., 2008). Nitrifying bacteria often oxidize soil $NH_4^+$ in excess, and



the resulting $NO_3^-$, which is prone to leaching, can lower soil nutrient content as well as contaminate groundwater, streams, rivers, and coastal waters, causing eutrophication (Lin et al., 2001; Beeckman et al., 2018). $NH_y$ directly falling onto natural waters is potentially toxic to aquatic life even in low concentrations, and can deteriorate marine biodiversity (Zhang and Liu, 1994; Shou et al., 2018).

70         The severity of the aforementioned consequences of excessive reactive nitrogen in the environment has called for better management of these compounds, including better monitoring and mitigation of agricultural $NH_3$. Various technologies for measuring ambient $NH_3$ have been deployed since the last century (Erisman et al., 2001; Fowler et al., 2009). One such example is flux towers that combine laser detectors with an eddy-covariance method to
provide on-site observations with finer time resolution (Sutton et al., 2008; Famulari et al., 2004; Ferrara et al., 2012; Zöll et al., 2016). Several national and international monitoring networks have also been established to obtain baseline ambient concentrations of $NH_y$, e.g., National Air Quality Monitoring Network in the Netherland (Buijsman et al., 1998), National Ammonia Monitoring Network (NAMN) in the UK (Sutton et al., 2005), National Atmospheric
Deposition Program (NADP) in the US (Puchalski et al., 2011), European Monitoring and Evaluation Programme (EMEP) in Europe (Fagerli and Aas, 2008), and the Acid Deposition Monitoring Network in East Asia (EANET; https://www.eanet.asia/), to determine the rates of $NH_4^+$ deposition and $NH_3$ emission. Over remote areas, airborne (e.g., Nowak et al. (2010) and Leen et al. (2013)) and ship (e.g., Norman and Leck (2005) and Wentworth et al. (2016)) $NH_y$
observations that combine near-field remote sensing and onboard in-situ measurement platforms have also been employed to fill the measurement gap. In the recent decade, the space-based Infrared Atmospheric Sounding Interferometer (IASI) has been deployed to gauge atmospheric $NH_3$ concentration within air columns (Clarisse et al., 2009). This new ensemble of satellite observations offers significant progress to address previous observational
deficiencies and allows daily monitoring of global $NH_3$ distribution (Clarisse et al., 2010; Van Damme et al., 2014). Continued refinement in retrieval schemes and incorporation of machine-learning techniques have further improved the sensitivity and reliability of measured $NH_3$ concentrations (Van Damme et al., 2017). It enables the creation of near real-time high-resolution maps of atmospheric $NH_3$ and the possibility of pinpointing industrial and
agricultural emission hotspots with diameters smaller than 50 km (Van Damme et al., 2018).





Their works have provided valuable datasets not only for monitoring agricultural emissions but also for benchmarking and improving emission inventories and numerical models.

$NH_3$ emission inventories are generally compiled by surveyed activity data and empirical emission factors associated with primary sources including animal populations, synthetic nitrogen fertilizers, biomass burning, and natural sources. A 1º-by-1º inventory, which was among the first back then, estimated a global emission of 54 Tg-N $yr^{-1}$ for 1990, of which 34 Tg-N $yr^{-1}$ is agricultural, excluding field burning, and 2.4 Tg-N $yr^{-1}$ from natural soil (Bouwman et al., 1997). Since then, much effort has been put into refining the estimation of anthropogenic emissions. Recent inventories adjusted the estimated agricultural emission in 2000–2008 to 33–37 Tg-N $yr^{-1}$ (Sutton et al., 2013). One of the state-of-the-art inventories, the Emissions Database for Global Atmospheric Research (EDGAR) version 4.3.2, provides global anthropogenic emission estimates in 0.1º-by-0.1º resolution for the period 1970–2012 (Crippa et al., 2018). The accuracy of these inventories is not only affected by the integrity of the activity data surveyed, but also constrained by the suitability of emission factors. Simply adopting emission factors from other countries may result in biases because of regional differences in technologies, farming practices, climate, and soil conditions (Huang et al., 2012). This pitfall has motivated the development of other national and regional inventories in the US (e.g., US Environmental Protection Agency (2014)), China (e.g., Zhang et al. (2018)), and Europe (e.g., European Environment Agency (2013)). These emission inventories are useful tools for source apportionment and input data for forward models but as the $NH_3$ emissions are prescribed they do not respond to changes in, e.g., nitrogen deposition and meteorology, making them insufficient for models to represent the full dynamics of the $NH_y$ cycle.

The global $NH_y$ cycle has proven to be challenging to study because of the various feedback mechanisms within the Earth system. The reactive nature of $NH_3$ and the contribution of deposited $NH_4^+$ to the re-emission of $NH_3$ from natural and agricultural soils have created a convoluted relationship between emissions and deposition. $NH_4^+$ particles can be transported along with airflows and dispersed across a more extensive geographical range than the highly reactive gaseous $NH_3$. Such transport can introduce large heterogeneity in the spatial distribution of reactive nitrogen, rendering it not only a local but pan-regional problem (Willem Asman; Mark A. Sutton, 1998). Moreover, $NH_3$ volatilization is a temperature-dependent process while the presence of atmospheric $NH_3$ affects the composition of aerosols and their



radiative forcing, thus in turn modifying the Earth's surface energy budget (Ansari and Pandis, 1998).

In this study, we hence aim to enable modeling of the land-atmosphere bidirectional exchange of $NH_y$, so that we can quantify the dynamically evolving $NH_y$ cycle and feedback mechanisms associated with it under a changing environment. We employed the Community Earth System Model version 2 (CESM2), which has state-of-the-art model components representing the land, atmosphere, sea ice, and oceans. These sub-models can run independently or in various coupled configurations (Hurrell et al., 2013). Many studies have
employed CESM for studying processes in both the atmospheric and terrestrial nitrogen cycles, e.g., $NO_x$ and $N_2O$ emission (Saikawa et al., 2013, 2014; Zhao et al., 2017), denitrification and nitrate leaching (Nevison et al., 2016), crop nitrogen uptake (Levis et al., 2018), and reactive nitrogen input to ecosystem associated with synthetic and manure fertilizers (Riddick et al., 2016). Yet, these studies did not consider the dynamic bidirectional transfer of $NH_3$ and $NH_4^+$
between the land and atmosphere. To add the dynamic cycle of $NH_y$ back to CESM2, we adopted a process-based approach to parameterize $NH_3$ emission from cropland soils, which is different from the "voltage-resistance" models (Riddick et al., 2016; Vira et al., 2020). We also developed a prognostic parameterization for canopy capture of $NH_3$, instead of using a fixed generic value (e.g., one constant canopy reduction factor for all plants as used in many other
studies (Riddick et al., 2016; Bouwman et al., 1997). Implementing these new schemes in the Community Land Model version (CLM5) (Lawrence et al., 2019), we could then estimate the emission associated with fertilizer use and perform fully coupled simulations with the Community Atmosphere Model version 4 with Chemistry (CAM4-Chem) (Lamarque et al., 2012) that allow two-way exchange of $NH_y$ bridged by online emission and deposition to
understand the subsequent effects on aerosol formation, climate, terrestrial ecosystems, and crop growth. We also compared our results with available emission inventories to evaluate model accuracy and uncertainty. This paper demonstrates a framework to unfold the complicated interactions between fertilizer use, $NH_3$ emission, aerosol formation, climate, terrestrial ecosystems, and crop production.




## 2. Methods

### 2.1 Community Earth System Model

We introduced new functionalities into CESM2 to enable the simulation of a coupled land-atmosphere nitrogen cycle, and to further investigate the impacts of fertilizer-induced $NH_3$ emission on atmospheric composition, terrestrial biogeochemistry, and climate change. In particular, we implemented into CLM5 new parameterization schemes to quantify $NH_3$ volatilized from soil due to fertilizer application and captured by plant canopies. We further bridged CLM5 and CAM4-chem to enable two-way exchange of soil $NH_3$ emission and deposition of $NH_4^+$ to model a fully coupled, prognostic land-atmospheric $NH_y$ cycle.

Our model development was based on CLM5 with active biogeochemical cycles and crop sub-model (CLM5-BGC-Crop or CLM5 in short), which represents terrestrial carbon and nitrogen cycling with prognostic vegetation and crop growth (Lawrence et al., 2019). The model uses a sub-grid hierarchy (from grid cells, land units, columns, to patches) to capture the biogeophysical and biogeochemical differences between various land types within a model grid cell. In particular, CLM5 handles natural soil and croplands differently: multiple natural vegetation patches are configured to occupy a single unmanaged soil column sharing a single pool of nutrients while each crop patch has a dedicated column. Such setting allows no resource competition between natural vegetation and crops, nor among crops (Drewniak et al., 2013). There are 16 types of natural vegetation (including bare ground) and eight active crops (temperate soybean, tropical soybean, temperate corn, tropical corn, spring wheat, cotton, rice, and sugarcane) in this model. Vegetation and crops are represented by plant functional types (PFTs), each having specific ecophysiological, phenological and biogeochemical parameters (Levis et al., 2018). Default PFT distribution of natural vegetation and crops are derived from satellite observations (e.g., MODIS) and agricultural census data (Lawrence and Chase, 2007; Portmann et al., 2010). The beginning of plant growth stages (seedling, leaf emerging, and grain filling), as well as crop sowing dates and planting durations, are controlled by cumulative warm-enough hours at the beginning of spring. Crops obtain nutrients from the soil mineral nitrogen pool, which is supplied by nitrogen deposition and fertilization. Fertilizer is applied to each patch for 20 consecutive days evenly when the crops enter the leaf emergence phase. Crops are harvested once they reach maturity or predefined maximum growing days (typically 150–165 days) (Lawrence et al., 2020).



## 2.2 Soil ammonia emission and canopy capture

**Figure 1** summarizes the primary pathways of the terrestrial nitrogen cycle in CLM5. The model tracks nitrogen content in soil, plant, and organic matter as an array of separate nitrogen pools, and biogeochemical processes as exchange fluxes of nitrogen between these pools. Soil mineral nitrogen, $NH_4^+$, and $NO_3^-$ are competed for among plant uptake, microbial immobilization, nitrification, and denitrification, based on the relative demand from each process. Release of nitrous oxide ($N_2O$) and $NO_x$ as byproducts of nitrification and denitrification and leaching of soil nitrate also deplete soil $NH_4^+$ and $NO_3^-$, which can be replenished by fertilization and deposition of atmospheric $NH_y$ and $NO_x$. The deposition rates were prescribed in the default configuration and dynamically computed by CAM4-chem in our version. Other sources of soil mineral nitrogen include biological fixation by microbes or soybean and decomposition of plant litter and soil organic matter. Our proposed $NH_3$ emission scheme was derived from another standalone biogeochemical model, the DeNitrification-DeComposition (DNDC) model version 9.5 (Li et al., 2012), which has been used for studying agricultural $NH_3$ emission (Balasubramanian et al., 2015; Zhang and Niu, 2016; Balasubramanian et al., 2017).





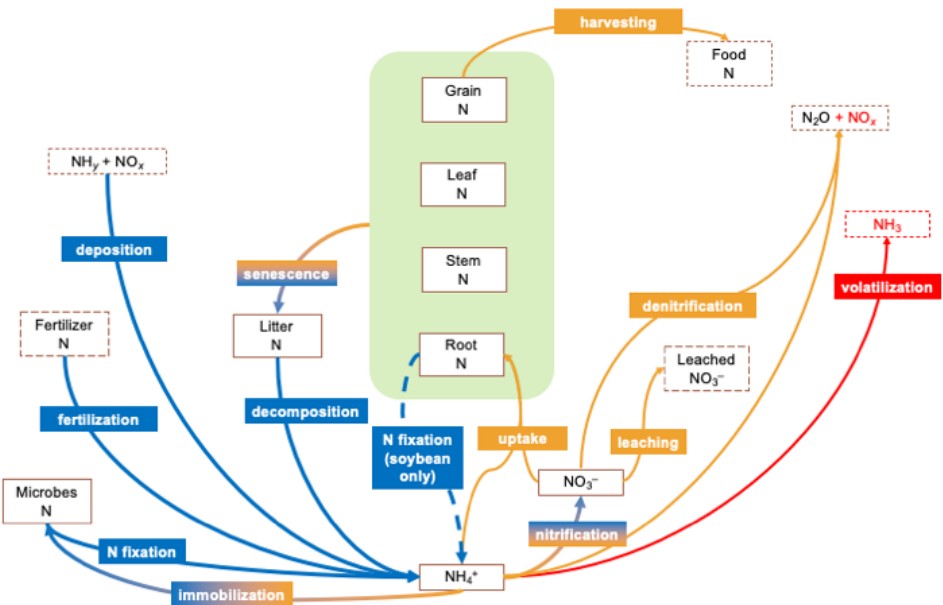

**Figure 1.** Major pathways modeled by CLM5 nitrogen (N) cycle. Blue arrows indicate N entering the soil N nitrogen pool while orange arrows are for leaving. The default model tracks only N pools in boxes enclosed by solid lines, but not those with dashed lines. N contents in crop tissues are modeled as pools inside the green regions. The red arrow indicates the missing pathway of NH$_3$ volatilization in the default model.

In our scheme, NH$_3$ volatilization is considered as a multistage process, and we estimated a potential soil NH$_3$ emission rate of each soil layer in each column or patch as:

$$\frac{\mathrm{d}[NH_{3\,(g)}]}{\mathrm{d}t}\bigg|_{soil} = [NH^+_{4\,(soil)}](1 - f_{ads})f_{dis}f_{vol}(\frac{1}{\Delta t}) \tag{1}$$

where $[NH_4^+{}_{(soil)}]$ (g-N m$^{-2}$) is the amount of soil NH$_4^+$; $\Delta t$ is model the time step size in CLM5 (default = 1800 s); $f_{ads}$ accounts for the portion of NH$_4^+$ adsorbed onto the surface of soil matrix; $f_{dis}$ is the fraction of the non-adsorbed NH$_4^+$ that dissociated into aqueous NH$_3$, and; $f_{vol}$ is the fraction of aqueous NH$_3$ volatilized as gaseous NH$_3$. The adsorbed fraction $f_{ads}$ is given by:

$$f_{ads} = 0.99(7.2733f_{clay}^3 - 11.22f_{clay}^2 + 5.7198f_{clay} + 0.0263) \tag{2}$$

where $f_{clay}$ is soil clay fraction as prescribed by the CLM surface data (Bonan et al., 2002).





The non-adsorbed $NH_4^+$ dissociates reversibly into aqueous $NH_3$ and hydrogen ion

($NH_4^+{}_{(aq)} \rightleftharpoons NH_{3(aq)} + H^+$), and hence, $f_{dis}$ is determined by the following equations (Li et al., 2012):

$$f_{dis} = \frac{K_w}{K_a[H^+]} \tag{3}$$

$$K_w = 10^{0.08946+0.03605T_{soil}} \times 10^{-15} \tag{4}$$

$$K_a = (1.416 + 0.01357T_{soil}) \times 10^{-5} \tag{5}$$

$$[H^+] = 10^{-pH} \tag{6}$$

where $K_a$ (in mol $L^{-1}$) and $K_w$ (in mol $L^{-2}$) are dissociation constants for $NH_4^+/NH_3$ and hydrogen-/hydroxide-ion equilibria, respectively; $T_{soil}$ (in °C) is soil temperature; $[H^+]$ (in mol) is the concentration of aqueous hydrogen ion in the soil calculated from soil pH. CLM5 is currently not capable of calculating soil pH implicitly so we performed our simulations using

a constant pH of 6.5 for a more focused analysis. This pH value is consistent with the value used in the nitrification and denitrification schemes in CLM5 (Lawrence et al., 2019). We further evaluated the uncertainty induced by our choice of pH and presented the sensitivity test results in **Appendix**.

Lastly, we used this equation to calculate $f_{vol}$:

$$f_{vol} = \left(\frac{1.5s}{1+s}\right)\left(\frac{T_{soil}}{50+T_{soil}}\right)\left(\frac{l_{max}-l}{l_{max}}\right) \tag{7}$$

where $s$ (in m $s^{-1}$) is surface wind speed; $T_{soil}$ (in °C) is soil temperature; $l$ and $l_{max}$ (both in m) are the depth of each particular soil layer and the maximum depth of a soil column, respectively. The actual soil $NH_3$ emission rate is then determined by the lower of the potential soil $NH_3$ emission rate or the available soil $NH_4^+$.

If vegetation is present above the soil, some of the emitted $NH_3$ can be retained by the plant canopy, which is known to be related to the adsorption of hydrophilic $NH_3$ onto the leaf surface and molecular diffusion via the leaf stomata (Van Hove et al., 1987). Some studies represented the amount of captured $NH_3$ using constant scaling factors (e.g., 0.6 for all vegetation in Riddick et al. (2016), 0.8, 0.5, and 0.2 for tropical rainforests, other forests, and





all other vegetation types, respectively, in Bouwman et al. (1997)). Here, we calculated the flux
        of $NH_3$ captured by canopy following the equation used in DNDC that accounts for the change
        in in-canopy $NH_3$ concentration, deposition velocity of ammonia, leaf area index (LAI), and
        air moisture (Institute for the Study of Earth, Oceans, and Space, University of New Hampshire,
        2017). To include the dynamic growth of crop canopy, we further adopted the canopy height

adjustment factor employed by the Community Multiscale Air Quality (CMAQ) regional
        chemical transport model (Pleim et al., 2013). The change in in-canopy $NH_3$ concentration is
        thus:

$$\frac{d[NH_{3\,(g)}]}{dt}\bigg|_{canopy} = \left(\frac{d[NH_{3\,(g)}]}{dt}\bigg|_{soil}\right) f_{can} \tag{8}$$

$$f_{can} = \frac{L}{s_{10}} v_c \varphi_c b(h_{top} - h_{bot}) \tag{9}$$

where $L$ is one-sided snow-free LAI, $s_{10}$ is the wind-speed (in m s$^{-1}$) at 10-m height, $v_c$ is the
        deposition velocity of $NH_3$ (0.05 m s$^{-1}$ as in DNDC), $\varphi_c$ is in-canopy relative humidity, $b$ is a
        correction factor for the effect of canopy thickness (14 m$^{-1}$ is used here as suggested by Pleim
        et al. (2013)), and $h_{top}$ and $h_{bot}$ are heights of canopy top and bottom (both in m), respectively.
        Except for $v_c$ and $b$, all variables are calculated within CLM5 (see Lawrence et al. (2019) for

detailed calculation methods). These two equations estimate the concentration of $NH_3$ exposed
        to plant canopy under a given soil emission rate at each time step: dividing soil $NH_3$ emission
        rate by $s_{10}$ gives an approximate in-canopy $NH_3$ concentration, and multiplying the latter with
        $v_c$ and $L$ produces an estimated quantity of $NH_3$ retained by the canopy. The last three terms
        account for the influence of in-canopy moisture and canopy thickness on the effectiveness of

canopy capturing. We subtracted this amount of captured $NH_3$ from the soil emission to obtain
        the quantity of $NH_3$ escaping to the above-canopy atmosphere, which was then used as input
        data to drive chemistry calculations in CAM4-chem. The captured $NH_3$ can re-enter the soil
        surface along with water throughfall or be metabolized by the plants if it diffuses into the leaf
        tissues (Hutchinson et al., 1972). Since the detailed mechanisms are still uncertain and beyond

the focus of this study, we decided to assume that all captured $NH_3$ returns to the soil directly
        as $NH_4^+$ and will discuss how it will affect our analysis in **Conclusions**.



**2.3 Emissions of other reactive nitrogen compounds**

In addition to the NH$_3$ schemes, we also incorporated new equations to calculate NO$_x$ released

as by-products of nitrification and denitrification. The original CLM5 estimates the amount of N$_2$O leakage during nitrification by applying a constant scaling factor to the nitrification rate (Li et al., 2000) while that from denitrification is variable and evaluated by the DayCent approach (Del Grosso et al., 2000). Building on the work of previous studies (Parton et al., 2001, 2004; Zhao et al., 2017), we computed a ratio of NO$_x$ to N$_2$O to account for the leaking

of the former during nitrification and denitrification using the following equations:

$$NO_x : N_2O = 15.2 + \frac{35.5 \tan^{-1}[0.68\pi(10D_r - 1.86)]}{\pi} \tag{10}$$

where $D_r$ is the relative gas diffusivity in soil vs. in air and is calculated as a function of air-filled pore space (AFPS) of soil (Davidson and Trumbore, 1995):

$$D_r = 0.209 AFPS^{\frac{4}{3}} \tag{11}$$

$$AFPS = 1 - \frac{\theta_V}{\theta_{V,sat}} \tag{12}$$

where $\theta_V$ and $\theta_{V,sat}$ are instantaneous and saturated volumetric soil water content (in m$^3$ m$^{-3}$), respectively.

In addition, we added back the 20% of microbial mineralized nitrogen to the nitrification rate, which was missing in the previous versions of CLM, following the DayCent

approach (Parton et al., 2001). We also applied a temperature factor to correct the overestimation of NO$_x$ emission at high latitudes as suggested in some previous studies (Xu and Prentice, 2008; Zhao et al., 2017):

$$f_T = \min\left(1, e^{308.56\left(\frac{1}{68.02} - \frac{1}{T_{soil} + 46.02}\right)}\right) \tag{13}$$

where $T_{soil}$ is soil temperature measured in Kelvin (K) here.






### 2.4 Simulations of the land-atmosphere NH$_y$ cycle

For the atmospheric component, we employed CAM4-chem, with chemistry based on the tropospheric chemistry mechanism of MOZART-4 (Emmons et al., 2010). CAM4-chem employs a bulk aerosol approach and predicts the formation of PM$_{2.5}$ components including

300 SO$_4^{2-}$, NO$_3^-$, and NH$_4^+$, where the injection rates of precursors – sulfur dioxide (SO$_2$), NO$_x$, and NH$_3$ – are prescribed by Coupled Model Intercomparison Project phase 6 (CMIP6) emission inventory for anthropogenic activities as well as some natural sources in the default configuration (Hoesly et al., 2018). Biogenic emissions are updated online from CLM5 using the Model of Emissions of Gases and Aerosols from Nature (MEGAN) version 2.1 (Guenther

305 et al., 2012). In our coupled simulations, we omitted the portion of NH$_3$ emission associated with synthetic fertilizer from the inventory input for CAM4-chem and replaced it with our online simulated emission rates from CLM5. Atmospheric NH$_3$ and NH$_4^+$ formed sequentially return to CLM5 through deposition.

 Dry deposition in CAM4-chem is handled using the resistance approach (Wesely, 1989;

310 Emmons et al., 2010). For NH$_3$ vapor, the model calculates the aerodynamic and the boundary-layer resistance based on the online atmospheric dynamics, while the surface resistance over land is determined according to the online CLM5 surface variables, e.g., canopy height and LAI, as well as species-specific reactivity factor for oxidation and effective Henry's Law coefficients. For particle-phase NH$_4^+$, the aerodynamic resistance is the same as that of NH$_3$,

315 but the boundary-layer and surface resistances are replaced by a single resistance term that depends on the surface friction velocity. The deposition velocities of NH$_3$ and NH$_4^+$ are the reciprocal of the sum of their corresponding resistance terms, and their deposition rates are the product of their deposition velocities and concentrations. Wet deposition in CAM4-chem follows the Neu and Prather (2012) scheme, which assumes a first-order loss of chemicals due

320 to in-cloud and below-cloud scavenging processes. The wet deposition rates of NH$_3$ and NH$_4^+$ are the products of their concentration, their loss frequencies (based on their Henry's Law coefficients), and the fraction of the grid box subject to scavenging (e.g., cloudy or raining). These NH$_y$ deposition fluxes then become the input to CLM5 as soil NH$_4^+$ (Lawrence et al., 2020).

325 Recent studies on NH$_3$ emission using CESM2 (e.g., Riddick et al. (2016) and Vira et al. (2020)) focused only on the one-way land-to-atmosphere flux of NH$_3$ while neglecting the





enhancing effect of nitrogen deposition on NH$_3$ emission. By coupling CLM5 and CAM4-chem, we allowed, for the first time, the model land-atmosphere NH$_y$ cycle to evolve in response to any changes in the bidirectional exchange of NH$_3$ and NH$_4^+$ via online emission

and deposition. It also makes our method more suitable than a one-way model for studying the feedback effects of future changes in climate and agricultural activities on the biogeochemical cycles.

**Table 1** provides configuration details of our experiments. All simulations were run for 10 years using year-2000 initial conditions with the corresponding land cover data. The first

five years of outputs were used as model spin-up, and thus, our analysis in the next section focused on the last five years of simulated results. The spatial resolution of our simulations was 1.9º by 2.5º horizontally with 32 layers of atmospheric layers from the surface to up to ~40 km, and 25 soil layers down to ~50 m below ground. Our analysis focused on the changes in fluxes of soil biogeochemical processes, the evolution of atmospheric NH$_3$, NH$_4^+$, and other SNA

aerosols, and the influence of the bidirectional NH$_y$ exchange on crop production. Fertilizer input was prescribed by crop type and country at the 2000-level based on Land-Use Harmonization (LUH2) fertilization rates (Hurtt et al., 2011). All fertilizers added were assumed synthetic and no manure was included.

**Table 1.** Details of simulation designs.

| Name | Fertilizer-induced NH$_3$ Emission | N-deposition | Aerosol-radiation Interaction |
|---|---|---|---|
| CAM4_CLM5 | This study | Dynamic | Enabled |
| CAM4_CLM5_NDEP | This study | Dynamic | Disabled |
| CAM4_CLM5_CLIM | This study | From [CAM4_CLM5] assuming 2000-level fertilization | Enabled |
| CAM4_CMIP6 | CMIP6 inventory | Dynamic | Enabled |

**2.5 Datasets for model validation**

We also compared our simulation results with various available observations and emission inventories. CLM5-modeled NH$_3$ emission was compared with multiple emission inventories including CMIP6, EDGAR, and the Magnitude And Seasonality of Agricultural Emissions for



NH$_3$ (MASAGE). CAM4-chem-simulated atmospheric NH$_3$ using CLM5 NH$_3$ and CMIP6

were compared against the satellite-derived IASI-NH$_3$ concentration field (gridded and reported in Van Damme et al. (2018). Details of these datasets are tabulated in **Table 2**. The datasets were regridded to match our model resolution of 1.9º by 2.5º using bilinear interpolation.

**Table 2.** Details of observations and emission inventories used in this study for model comparison and validation.

| Name | Coverage | Resolution | Period of data | Data type: sources extracted for model comparison |
|---|---|---|---|---|
| MASAGE (Paulot et al., 2014) | Global | 0.5º-by-0.5º Monthly mean | 2006 | Emission inventory: NH$_3$ emission from agricultural soil associated with synthetic fertilizers for crops |
| EDGAR (Crippa et al., 2018) | Global | 0.1º-by-0.1º Monthly mean | 2010 | Emission inventory: NH$_3$ emission from agricultural soil with fertilizer and manure application |
| CMIP6 (Hoesly et al., 2018) | Global | 0.01º-by-0.01º Monthly mean | 2006–2015 | Emission inventory: NH$_3$ emission from agricultural soil with fertilizer and manure application |
| IASI (Van Damme et al., 2018) | Global | 0.01º-by-0.01º Annual mean | 2008–2016 | Satellite-based measurement: Column NH$_3$ density |

**3. Results**

**3.1 Fertilizer-induced NH$_3$ emission**

We first evaluated the fertilizer-induced NH$_3$ emission simulated by the fully coupled land-atmosphere simulation, [CAM4_CLM5]. **Figure 2** shows the annual-total global NH$_3$ emission at above-canopy level from different land types averaged over the last five years of simulation.

We also compared our NH$_3$ emission with inventory estimates reported by CMIP6 (Hoesly et al., 2018), EDGAR v4.3.2 (Crippa et al., 2018), and MASAGE (Paulot et al., 2014). We extracted the monthly fertilizer-induced NH$_3$ emission estimates from MASAGE, and assumed that one-third of the total agricultural NH$_3$ emission reported by CMIP6 and EDGAR are fertilizer-associated, which is consistent with the apportionment reported in previous studies

and environmental reports (Paulot et al., 2014; Riddick et al., 2016; National Oceanic and Atmospheric Administration, 2000; European Environment Agency, 2010; Gu et al., 2012; Paulot et al., 2015; Zheng et al., 2017).



A grid cell-by-grid cell model-inventory spatial comparison of the annual-total $NH_3$ emission rates was conducted by computing Pearson's correlation coefficients ($R$) and slopes
($\beta$) of linear regression using the reduced major axis method as well as normalized mean biases (NMB; $\Sigma(M_i - O_i)/\Sigma(O_i)$, where $M_i$ and $O_i$ are simulated and inventory $NH_3$ emission in each grid cell) and mean fractional biases (MFB; $2\Sigma[(M_i - O_i)/(M_i + O_i)]/N$, where $N$ is the number of grid cell). A summary of these statistics is shown in **Fig. 2(a)**. We also computed the $R$ values between the monthly emission rates computed by CLM5 (as in the fully coupled case, [CAM4_CLM5]) and each inventory for each grid cell (see **Fig. 2(d)**, **(f)**, and **(h)**). We
highlighted with overlaying black dots the grid cells with high coefficients of determination, which are statistically significant (i.e., $R^2 > 0.5$ and $p < 0.05$), indicating where our simulation can reproduce more than half of the variability of the inventory estimates.

Globally, CLM5 estimates that the annual-total fertilizer-induced $NH_3$ emission
reaches 11 Tg-N yr$^{-1}$, which is close to the 12 Tg-N yr$^{-1}$ and 11 Tg-N yr$^{-1}$ reported by the two similar studies, Riddick et al. (2016) and Vira et al. (2020), respectively. Our estimate is slightly higher than all three inventories, which are 10 Tg-N yr$^{-1}$ for CMIP6 and EDGAR, and 9.1 Tg-N yr$^{-1}$ for MASAGE. The global $R$ values are positive and lie within 0.5–0.6 across all inventories, indicating a fairly good correlation between CLM5 and all three inventories,
especially CMIP6 and MASAGE. Systematic model high-biases are implied by the greater-than-unity $\beta$ values. Such high-biases are within acceptable levels as the NMB values are smaller than 200% and the magnitudes of MFB values are within 50% (Boylan and Russell, 2006).

Top food-producing countries are responsible for a major portion of the fertilizer-
induced $NH_3$ emission: 26% of CLM5 global total was from India, 17% from the US 17%, and 9.7% from China. Emission hotspots are found close to their cropping regions in the model and the inventories, but their spatial gradients are different. In India, CLM5 shows more concentrated emission sources over the northern regions, resulting in higher local emission rates than the inventories. This distribution pattern resembles the India's north-higher south-
lower fertilization gradient adopted by the model. In contrast, CMIP6 estimates a more evenly distributed emission spatial pattern over India, and higher emission rates over the southern regions. EDGAR and MASAGE show a spatial gradient of $NH_3$ emission decreasing from north to south. Such gradients may explain their low $R$ and high $\beta$ values against our revised





CLM5. Despite the spatial mismatch, model estimation over India still falls in the acceptable
range in terms of the regional NMB and MFB.

CLM5 estimates more intense emission hotspots in the US, which are located near the
"Corn Belt" of the central US and southern California. US emission rates by CLM5 are much
higher than the other three inventories, as seen in the difference maps in **Fig. 2**, as indicated by
the large $\beta$ (>2.5), and large regional NMB. Differences in the spatial distribution of $NH_3$
emission are also observed over China. CLM5 estimates that more $NH_3$ is emitted from central
and northeastern China, while the emission hotspots in CMIP6 and EDGAR are found in
northeastern China and those of MASAGE are located in eastern China. Such deviation may
be attributable to different fertilizer usage schedules used by CLM5 and other inventories. For
example, MASAGE considers multiple-type fertilizers including ammonium bicarbonate,
which is more prone to $NH_3$ loss than urea, and assumes a three-stage fertilization at sowing,
growth, and harvesting (Paulot et al., 2014). EDGAR also reported a high uncertainty (~97%)
of present-day $NH_3$ emissions in China due to incomplete information about the agricultural
sector (Crippa et al., 2018).







**Figure 2.** Fertilizer-induced NH$_3$ emission estimated by CLM5 and other emission inventories. Correlation analysis between CLM5-simulated annual-total emission and other inventories with regional breakdowns is summarized in panel **(a)**. Spatial distribution of annual-total fertilizer-induced NH$_3$ emission simulated by [CAM4_CLM5], and estimated by CMIP6, EDGAR, and MASAGE are illustrated in panels **(b)**, **(c)**, **(e)**, and **(g)**, respectively. Panels **(d)**, **(f)**, and **(h)** show the spatial distribution of differences in annual-total NH$_3$ between

CLM5 and CMIP6, EDGAR, and MASAGE, correspondingly. Overlaying black dots in the difference maps indicate grid-cells with a high statistically significant spatiotemporal correlation (i.e., $R^2 > 50\%$, $p < 0.05$) between CLM5 and the corresponding inventories. Color scales are saturated at respective values, and ranges of values are shown in the legend titles.

425       **Figure 3** shows the seasonality of NH$_3$ emission associated with artificial fertilizer in the Northern and Southern Hemisphere. CLM5 assumes each crop receives a specific amount of fertilizer (as soil NH$_4^+$) applied evenly for 20 consecutive days since leaf emergence. This soil NH$_4^+$ input speeds up plant uptake, microbial immobilization, nitrification, as well as NH$_3$ volatilization, explaining the Northern Hemisphere peaking in emission in April and May and

Southern Hemisphere peaking in October, overlapping with the regional cropping seasons. All inventories show springtime peaks in each hemisphere, but the peak of EDGAR always leads the others by a month. CMIP6 has multiple peaks (two in the Northern Hemisphere and three in the Southern Hemisphere). These deviations exist mainly because of the differences in planting schedule and duration of fertilization used by the inventories. The higher CLM5 peaks

are consistent with the systematic overestimation discussed above. NH$_3$ emission returns to "background" levels when it is not in the planting seasons. EDGAR and CMIP6 have higher background levels because the original estimates accounted for not only synthetic fertilizer but also manure application (for both) and management (for CMIP6 only), which are not necessarily in phase with the cropping seasons (Huijsmans et al., 2018).

440       We concluded our model-inventory comparison by computing the correlation of monthly NH$_3$ emission rates in each model grid cell (see **Fig. 2(d)**, **(f)**, and **(h)**). CLM5 can capture a large portion of emission hotspots of CMIP6 over the US, Europe, India, China, and South America. With MASAGE, our estimate shows good agreement over mid-range emission regions, in North America, South America, Europe, and Southern Africa. CLM5 differs the

most from EDGAR among the three inventories. The resemblance with CMIP6 and MASAGE indicates that our NH$_3$ scheme has allowed CLM5 to produce reasonable NH$_3$ emission inputs

for CAM4-chem simulations over most high to medium emission hotspots. It is also noteworthy that the magnitude and spatial distribution of $NH_3$ emission among inventories are also not consistent. Since environmental conditions control the rate of biological and chemical

processes that release $NH_3$, processes such as urea hydrolysis and $NH_4^+/NH_3$ equilibrium can induce further inventory uncertainties (Hoesly et al., 2018). Inter-inventory uncertainties are also attributable to the choice of global and/or regional emission factors, which is crucial to reflect different agricultural procedures across the world, such as fertilization methods and fertilizer types, but not always well represented in global inventories (Paulot and Jacob, 2014;

Riddick et al., 2016; Zhang et al., 2018).

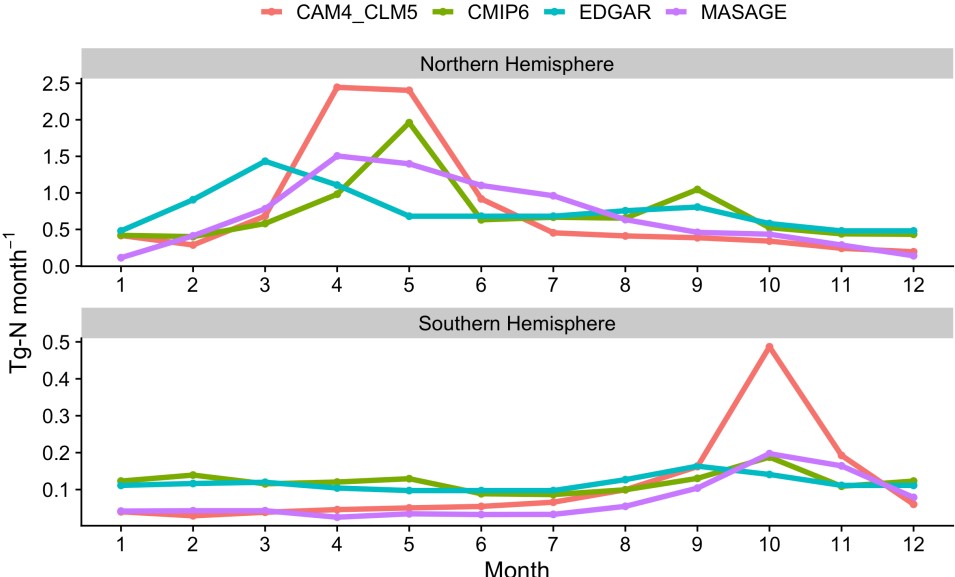

**Figure 3.** Monthly $NH_3$ emission associated with synthetic fertilizer use in the Northern and South Hemisphere estimated by CLM5, CMIP6, EDGAR, and MASAGE.


## 3.2 Atmospheric $NH_3$ concentration

Previous studies have evaluated the reactive nitrogen processes in CESM against satellite and ground observations, e.g., depositional fluxes of $NH_3$ and $NH_4^+$ (Lamarque et al., 2013), and



ground concentration of gaseous NH$_3$ (Hu et al., 2015). Here, we estimated the CAM4-chem
simulated annual-mean atmospheric NH$_3$ using two different inputs of fertilizer-induced
emissions simulated by our revised CLM5, and prescribed from CMIP6, which are aliased as
[CAM4_CLM5] and [CAM4_CMIP6], respectively. **Figure 4** shows these results, aggregated
to column total NH$_3$, alongside the 8-year annual-average IASI satellite retrievals.

Both simulations can capture the high concentration zones observed by IASI over the
US, South America, and Western Europe, with global and regional $R$ values > 0.8, indicating
a good correlation between the modeled results and observations. However, both
[CAM4_CLM5] and [CAM4_CMIP6] NH$_3$ are generally lower than IASI. Over the US and
India, [CAM4_CLM5] estimates lower NH$_3$ concentration than that from IASI (regional $\beta$ =
0.5–0.8), where CLM5 estimates high emission rates. Even more significant underestimation
is seen in [CAM4_CMIP6] (regional $\beta$ = 0.2–0.5), except in China, where lower emission rates
are predicted by [CAM4_CLM5] (**Fig. 2(d)**). The magnitudes of NMB and MFB of
[CAM4_CMIP6] are also more negative than [CAM4_CLM5], reflecting that using CLM5 as
NH$_3$ emission input reduces the model NH$_3$ underestimation of CAM4-chem with the default
CMIP6 inventory.

Mild differences are seen in North America and northeastern China, which are both
intense agricultural regions; the discrepancies are likely attributable to the mismatch in crop
growth map between CLM5 and the real world. Larger differences are shown over India and
Western Europe, indicating the low-biases in the model of emission from tropical biomass
burning regions (Whitburn et al., 2017; Van Damme et al., 2018).

We further compared the NH$_3$ burden of a run with online NH$_3$ emission and prescribed
N deposition, i.e., [CAM4_CLM5_CLIM], against IASI to examine whether enabling the
online bidirectional exchange can improve the estimation of NH$_3$ in CLM5. Turning on
nitrogen deposition, the [CAM4_CLM5]-estimated global-total NH$_3$ emission is 0.6% higher
than [CAM4_CLM5_CLIM]. Regional changes in NH$_3$ emission are not uniform (**Fig. S1**).
The most prominent changes are found in Asia (+0.09 Tg-N yr$^{-1}$ or +1.6%), and South America
(–0.09 Tg-N yr$^{-1}$ or –8.0%). When compared with IASI, [CAM4_CLM5] has closer-to-one $\beta$
and closer-to-zero NMB and MFB than [CAM4_CLM5_CLIM], indicating that the fully
coupled N-cycle could reduce model low-bias (**Fig. S2**).


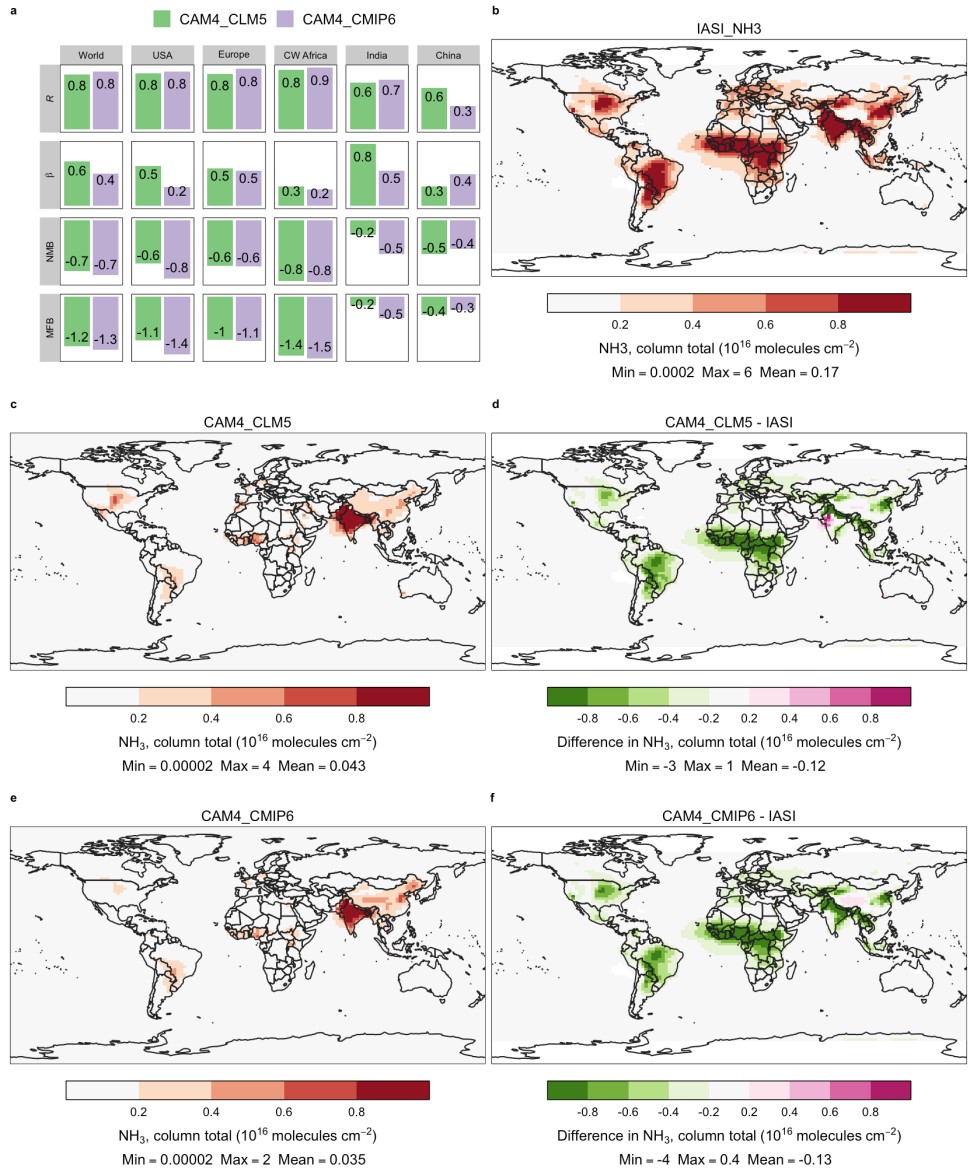

**Figure 4.** Annual-mean atmospheric NH₃ estimated by CAM4-chem with online CLM5 simulation and CMIP6 emission inventory as inputs of fertilizer-induced NH₃ emission, which are aliased as [CAM4_CLM5] and [CAM4_CMIP6], respectively. Panel **(a)** summarizes the correlation analysis between the two simulations and the IASI satellite retrievals. Panels **(b)**, **(c)**, and **(e)** show the column NH₃ concentration of IASI and the two cases correspondingly. Panels **(d)**, and **(f)** show concentration differences between each case and the IASI observations. Color scales are saturated at respective values, and ranges of values are shown in the legend titles.



### 3.3 When fertilizer application is increased by 30% – a case study to reveal the importance of nitrogen deposition and aerosol-climate effect on NH₃ emission and grain production in a future scenario

505 Fertilizer use is predicted to increase by >30% of 2000-level to boost grain production to meet the fast-growing food demand by 2050 (FAO, 2007). Such injection of soil nitrogen will not only enhance soil $NH_3$ emission, but also alter atmospheric $NH_4^+$ formation and its subsequent climate effects and deposition, which will induce secondary impacts on crop growth and $NH_3$ re-emission. Here, we used the modified CLM5 and CAM4-chem to attribute such secondary

510 impacts to nitrogen deposition and aerosol-climate effect. We performed this case study by scaling up the amounts of fertilizer application by 30% globally as input to the simulations detailed in **Table 1**.

 [CAM4_CLM5] encapsulates the full functionality of our implementation, i.e., that CAM4-chem receives the online CLM5 $NH_3$ emission rates as input to predict atmospheric

515 $NH_3$ concentration, the subsequent formation of ammonium salts, and the corresponding instantaneous aerosol radiative effect on climate, whilst CLM5 obtains the online CAM4-chem dry and wet deposition rates of $NH_y$ and $NO_x$ to calculate the addition of soil $NH_4^+$ via deposition. Such deposition will eventually enrich soil fertility and fuel the re-emission of soil $NH_3$ while the aforementioned aerosol radiative effect can cool the Earth's surface and

520 suppress $NH_3$ volatilization. [CAM4_CLM5_CLIM] was prescribed with constant nitrogen deposition fluxes so that we could quantify the instantaneous aerosol radiative effects. Similarly, [CAM4_CLM5_NDEP] was configured such that addition/reduction of atmospheric $NH_3$ would not induce changes in aerosol-climate interactions, so that we could isolate the impact of $NH_y$ deposition on $NH_3$ emission and crop growth.

525 **Figure 5** shows the changes in annual-total fertilizer-induced $NH_3$ emission estimated by these simulations when the global fertilizer use rises to 130% of the 2000 level. The fully coupled case, [CAM4_CLM5], estimates that the global emission will rise by 2.4 Tg-N yr⁻¹ or 23% more of fertilizer-associated $NH_3$ emission than the baseline case, i.e., [CAM4_CLM5] with 2000-level fertilization (see **Fig. 2(b)**). The nonlinear increase in emission relative to the

530 fertilizer increment is expected because of the handling of fertilizer in CLM5 – fertilizer is





added as $NH_4^+$ to soil evenly over 20 days when crop leaves begin to emerge, during which the $NH_4^+$ will be competed for among nitrification, plant uptake, microbial immobilization, as well as volatilization (as illustrated in **Fig. 1**). Our model modifications enabled CLM5 to simulate the dynamic competing processes depending on microbial activities, plant growth, and soil

microclimate, as well as depositional soil nitrogen input and overlying climatic changes.

With constant nitrogen deposition fluxes, [CAM4_CLM5_CLIM] shows that $NH_3$ emission could increase by 2.5 Tg-N yr$^{-1}$ from the baseline case under the future scenario. Comparing with [CAM4_CLM5], we find that, if aerosol-climate interactions are the only factor in play, $NH_3$ is reduced in the north-central US but increases in the south-central US (see

**Fig. S3**). Such regional gradient is attributable to the north-south difference in plant uptake – higher plant uptake leaves less $NH_3$ to undergo volatilization.

When the emitted $NH_3$ and the subsequently formed aerosols are not allowed to affect climate radiatively, changes in [CAM4_CLM5_NDEP] are solely driven by the changes in nitrogen deposition, and such an effect enhances global $NH_3$ by 2.7 Tg-N yr$^{-1}$. These

amplifying effects are most prominent in India, which CLM5 determines as an emission hotspot, driving a net increase by 1.6 Tg-N yr$^{-1}$ in Asia relative to the baseline. Overall, we observe that nitrogen deposition leads to higher magnitudes of positive changes in $NH_3$ emissions than aerosol climate effects in all regions, except in the US. The smaller increment in the US is likely due to the lower surface temperature over the high-emission Corn Belt region

(**Fig. S5**) that suppresses $NH_3$ volatilization.

Though the 30% addition of fertilizer use increases $NH_3$ emission in all cases, the changes in [CAM_CLM5] are not the simple mean of the two other cases, implying that the combined effect of nitrogen deposition and aerosol-climate interactions is nonlinear. For instance, the lower surface temperature over the US Corn Belt is associated with higher latent

heat flux (**Fig. S7**), which is likely a consequence of better vegetation growth driven by increased $NH_y$ deposition following higher $NH_3$ emissions. This fertilizing feedback effect via N deposition is not seen in [CAM4_CLM5_CLIM], which receives constant, prescribed N deposition fluxes. Such a nonlinear feedback effect highlights the importance of a land-atmosphere coupled model, as in CESM2, when considering the environmental and climate

impacts of agricultural $NH_3$.



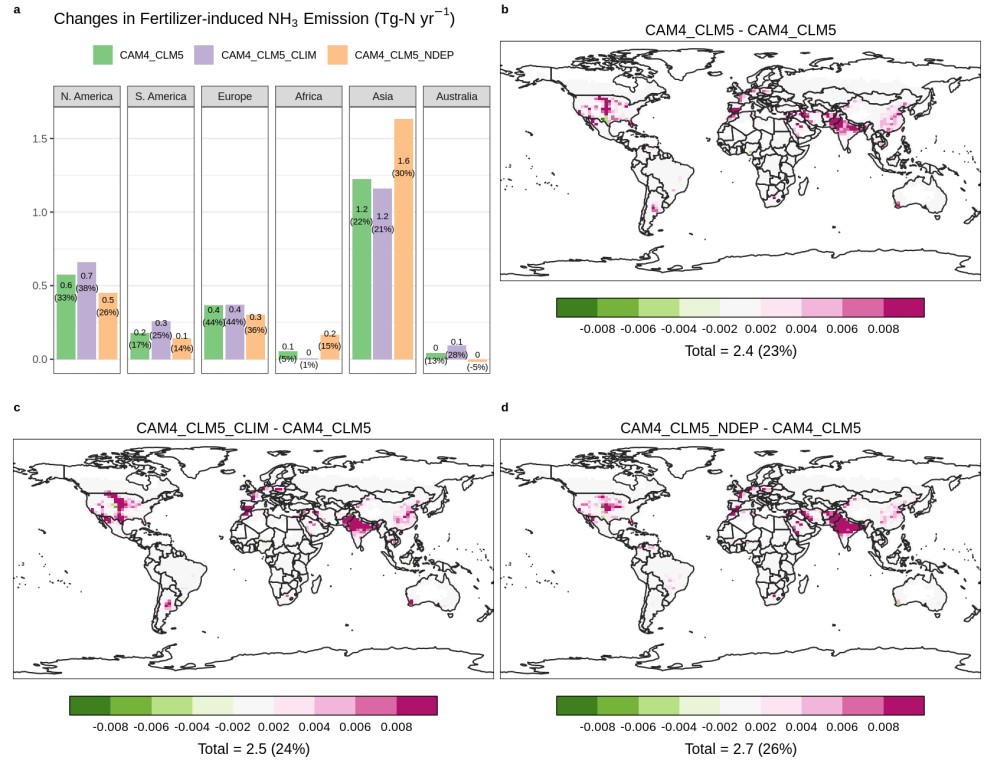

**Figure 5.** Changes in annual-total fertilizer-induced NH₃ emission after a 30% fertilization increase relative to [CAM4_CLM5] with 2000-level fertilization.

This nonlinearity is also exhibited in simulated grain production. CLM5 assumes a
harvest efficiency of 85% (Lawrence et al., 2020). [CAM4_CLM5] estimates that, with
nitrogen deposition and aerosol climate effect, 200 Tg yr⁻¹ more grain is produced than the
control (see **Fig. 6**). Such enhancement is found dominantly over the food-producing regions
over the US Corn Belt and Western Europe, resulting in net regional increases by 54 Tg yr⁻¹
and 108 Tg yr⁻¹ in North America and Europe, respectively. It is noteworthy that the combined
effect has a divergent impact and leads to production loss in South America, Africa, and
Australia. Looking into the two other cases, [CAM_CLM_NDEP] shows that nitrogen
deposition contributes to a large grain increase in Asia, North America, and Europe, which is
a direct benefit of the extra input of soil NH₄⁺ for crop growth improvement. Changes in
[CAM_CLM5_CLIM] indicate that the aerosol climate effect offsets the intensified
fertilization in virtually all continents, except in South America and Australia, which may be



attributable to the prolonged growing period and shortened grain filling stage. For example, as seen in **Fig. S6**, surface temperature in North America is lower in [CAM4_CLM5_NDEP] than [CAM4_CLM5], and such cooler surface is associated with higher yield increases over the region as shown in **Fig. S5**. These results show that the combined impacts of the two processes

not only vary nonlinearly but are also spatially uneven.

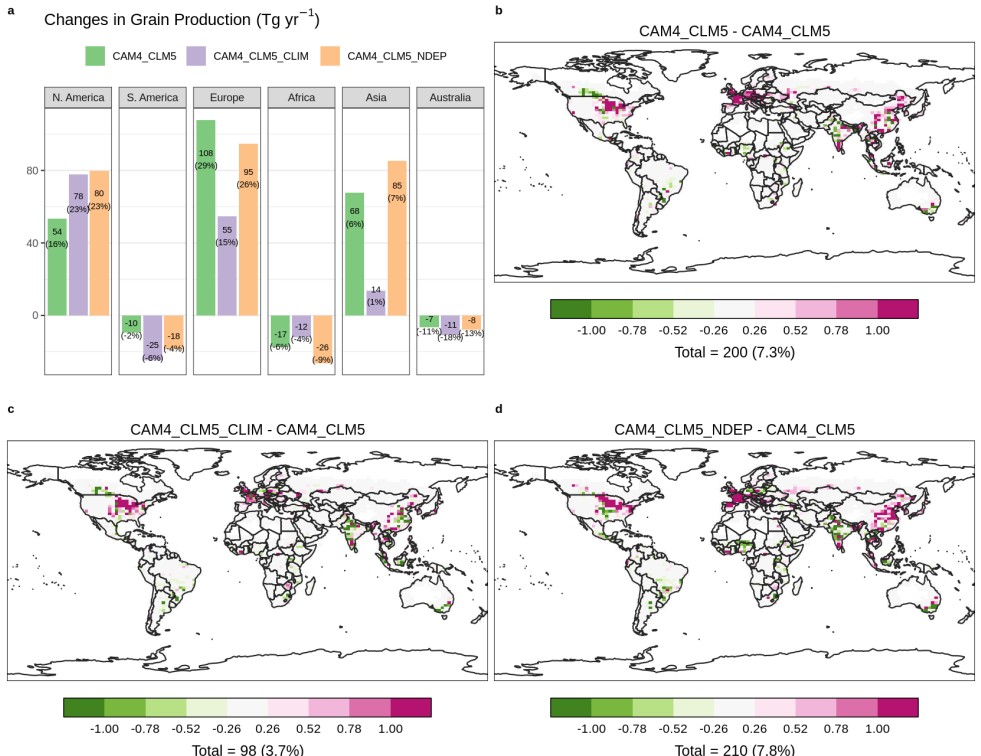

**Figure 6.** Same plots as **Fig. 5** but for annual-total grain production.

## 4. Conclusions

In this study, we implemented into the land and biogeochemical model, CLM5, new mechanistic schemes to better represent fertilizer-induced $NH_3$ emission from agricultural soil. Our modifications allowed CLM5 and CAM4-chem to dynamically exchange fluxes associated with reactive nitrogen deposition and $NH_3$ emission. These new features enabled CESM2 to





perform, for the first time, a more reliable estimation of soil $NH_3$ emission and atmospheric
$NH_3$ concentration under dynamic climate and environmental conditions. We verified that a
fully coupled simulation case, [CAM4_CLM5], produced an estimation of $NH_3$ emission that
agrees fairly well spatially and temporally with the emission inventories, MASAGE (Paulot et
al., 2014) and CMIP6 (Hoesly et al., 2018), especially over high-emission regions (see **Fig. 2**).
When compared to the IASI satellite observations (Van Damme et al., 2018), online $NH_3$
emission input in CLM5 reduces the low-biases exhibited in CAM4-chem estimation of
atmospheric $NH_3$ using the CMIP6 $NH_3$ emission inventory.

Our modifications also enabled us to understand how $NH_3$ emission influences aerosol
formation and aerosol radiative effect, and their secondary impacts on the re-emission of $NH_3$
and grain production. Our fully coupled simulation, [CAM4_CLM5], re-creates the spatial
distribution of the emission hotspots observed by satellite over intensive agricultural regions
including China, India, Europe, and the US (see **Fig. 4**). We estimated that the effect of nitrogen
deposition on $NH_3$ emission is +2.7 Tg-N yr$^{-1}$ globally, and is adjusted to +2.4 Tg-N yr$^{-1}$ when
the aerosol-climate effect is accounted for as well, compared with the baseline case (see **Fig.
5**).

This study demonstrates a modeling approach to estimate the climatic and
environmental sensitivity of $NH_3$ emission, with a focus on sources associated with synthetic
fertilizer only. Other primary sources of atmospheric $NH_3$ include manure management and
application (47%), ocean (16%), and biomass burning (11%). (Bouwman et al., 1997; Sutton
et al., 2013; Paulot et al., 2014, 2015). Unlike soil emission whereby the volatilization of $NH_3$
depends on a series of biogeochemical processes, emissions associated with manure
management from confined facilities, e.g., animal factories, are easier to track. To quantify
such emissions, one can collect activity data and emission factors from factory managers, and
install monitoring instruments at facility outlets (Bouwman et al., 1997; Paulot et al., 2014).
Manure usage for fertilizing croplands can be collected by surveying practices adopted by
farmers. Its associated $NH_3$ emission can be estimated using source-specific emission factors
and weather data, especially dominant factors such as air temperature, wind speed, and
humidity. Fire emission directly injects the reactive nitrogen into the atmosphere, and satellite
measurement is capable of capturing such short-term ammonia blooms (Van Damme et al.,
2017). We did not include manure application in our study due to the high uncertainty and data





insufficiency for validation. It is noteworthy that manure fertilizer is attributable up to ~25% of total $NH_3$ emission (Riddick et al., 2016) and hence shall warrant further research efforts in terms of its downstream impact on ecosystems via nitrogen deposition and aerosol radiative effect.

We also incorporated a prognostic parameterization for canopy capture of the emitted
$NH_3$, which is an improvement when compared to previous studies that assigned blanket reduction factors to all vegetated land types (Bouwman et al., 1997; Riddick et al., 2016; Vira et al., 2020). Despite such addition, our model still shows systematic high-biases, implying room for improvement, including further calibration of the canopy capture effects against field measurements. Another source of uncertainties stems from the model's initial soil $NH_4^+$ content,
which determines the potential emission rate of $NH_3$. The overestimation by CLM5 in this study may point to the more-fertile-than-reality soil conditions in the model, highlighting the need for a more realistic soil nitrogen map compiled by field surveys. We also note that such field surveys would also be useful to infer a soil pH map that constraints the uncertainty in simulations using a constant pH, like those reported in this study.

Our schemes simplified the fate of $NH_3$ captured by the canopy and assumed that such $NH_3$ is returned to the soil and becomes immediately accessible to plants, soil microbes, and bacteria, due to limited knowledge of the consequences of the canopy capturing process. A chamber study suggested that soybean can absorb up to 20 kg-N ha$^{-1}$ of $NH_3$ via leaf capturing (Hutchinson et al., 1972), which is a significant amount compared to average fertilizer use for
soybean of 25 kg-N ha$^{-1}$ in previous versions of CLM. On the other hand, concentrated $NH_3$ could damage leaf tissues if the contacting plant fails to metabolize or detoxify such a reactive gas in time (Nemitz et al., 2001). The remaining captured $NH_3$ on the leaf surface can return to the soil via throughfall, but its magnitude is difficult to measure. These unspecified processes may induce uncertainties in our simulations, especially for plant growth and soil $NH_4^+$ content.
This knowledge gap points to a demand for more field experiments to investigate the impacts of these processes.

FAO projects that fertilizer use will be increased by >30% (FAO, 2007) to boost food production to meet the fast-growing food demand by 2050. Such additional fertilizer injects mineral nitrogen into the soil that further fuels the volatilization of $NH_3$ spontaneously and
hence promotes the subsequent formation of aerosol particles. This study shows the nonlinear





impacts of nitrogen deposition and aerosol radiative effect on the environment. Thus, our work makes it possible to evaluate the intertwined consequences of such soaring use of fertilizer on $NH_3$ emission, atmospheric aerosols composition, and the corresponding aerosol-climate effect. Our results can provide scientific information to aid stakeholders in evaluating various global

and regional plans for mitigating climate change and safeguarding a sustainable environment.





## Appendix

### A. Canopy Capture Fraction

**Figure A1** shows the time series of the fractional amount of NH₃ captured by above-ground

crop biomass. Our scheme showed that crops do not capture as much NH₃ as natural vegetation

(~60–80% as estimated in our scheme) due to their smaller canopies, except for sugarcane and

temperate soybean, which can retain >30% of NH₃ emitted. CLM5 allows crops to be irrigated

or rainfed. We observed a general trend that irrigated crops retain more soil NH₃ emission than

rainfed ones, which can be explained by their higher in-canopy air humidity. Despite

instantaneous values varying with plant growth, our calculated canopy capture fractions are

close to those used in previous studies (Bouwman et al., 1997), i.e., 20% for other shrubs,

grasses, and crops.

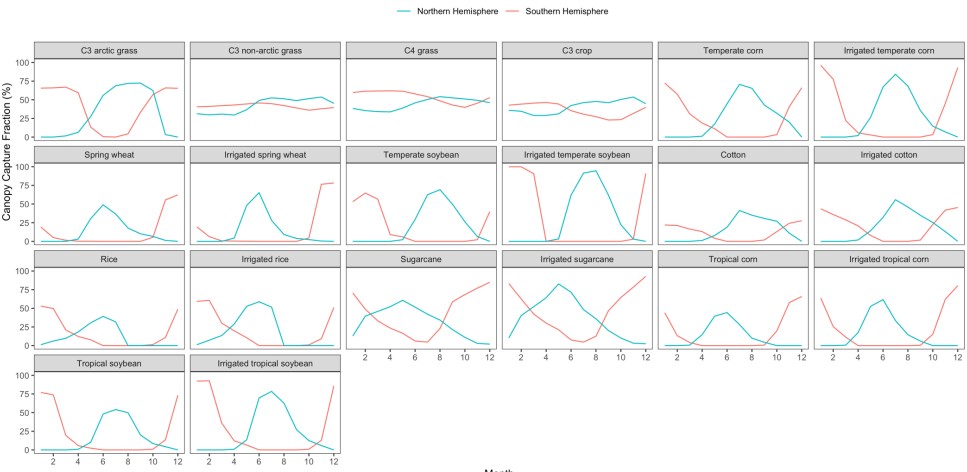

**Figure A1.** Monthly global-mean fractions of NH₃ captured (%) by each PFT canopy in each hemisphere.





**Table A1. Annual mean fraction of NH3 captured (%) by PFTs.**

| Crops | %NH$_3$ Captured |
|---|---|
| Temperate corn | 24.8 |
| Irrigated temperate corn | 27.1 |
| Spring wheat | 12.7 |
| Irrigated spring wheat | 14.5 |
| Temperate soybean | 19.9 |
| Irrigated temperate soybean | 30.1 |
| Cotton | 13.2 |
| Irrigated cotton | 19.7 |
| Rice | 14.3 |
| Irrigated rice | 19.0 |
| Sugarcane | 37.2 |
| Irrigated sugarcane | 41.2 |
| Tropical corn | 14.5 |
| Irrigated tropical corn | 18.6 |
| Tropical soybean | 19.3 |
| Irrigated tropical soybean | 25.4 |

**B. Sensitivity to Soil pH**

CLM5 does not have a built-in method to compute soil pH implicitly. Thus, in **Results**, we
used a constant global pH of 6.5, based on the implementation of the NH$_3$ volatilization scheme
in DNDC (Li et al., 2012), to avoid the uncertain and highly spatial varying soil acidity. We
tested the sensitivity of our simulated NH$_3$ emission by using the soil pH map from the
Harmonized World Soil Database (HWSD) v1.2 (Wieder, 2014) showed in **Fig. B1(c)**. This
soil pH dataset shows that the middle 50% of soil has pH values ranged from 5.4–7.0. When
comparing maps in **Fig. B1(b)** and **(c)**, we observed that the more alkaline the soil is, the more
soil emits NH$_3$. From our scheme (**Eq. (3)** in particular) for NH$_3$ volatilization, the emission
rate is of the order of $10^{pH}$:

$$\frac{d[NH_{3\,(g)}]}{dt}\bigg|_{soil} \sim O\left(\frac{K_w}{10^{-pH}K_a}\right) \sim O'(10^{pH}) \tag{B1}$$

**Figure B1(d)** also illustrates and confirms this exponential relation between NH$_3$
emission and pH, pointing to a demand for an implicit approach for calculating prognostic soil
alkalinity. Current version of CLM5 tracks only a few chemicals in soil, including NH$_4^+$, NO$_3^-$,

and methane ($CH_4$), making the calculation of bulk soil pH difficult. Future models shall include crucial chemicals and processes, as characterized by experimental studies, that affect soil pH.

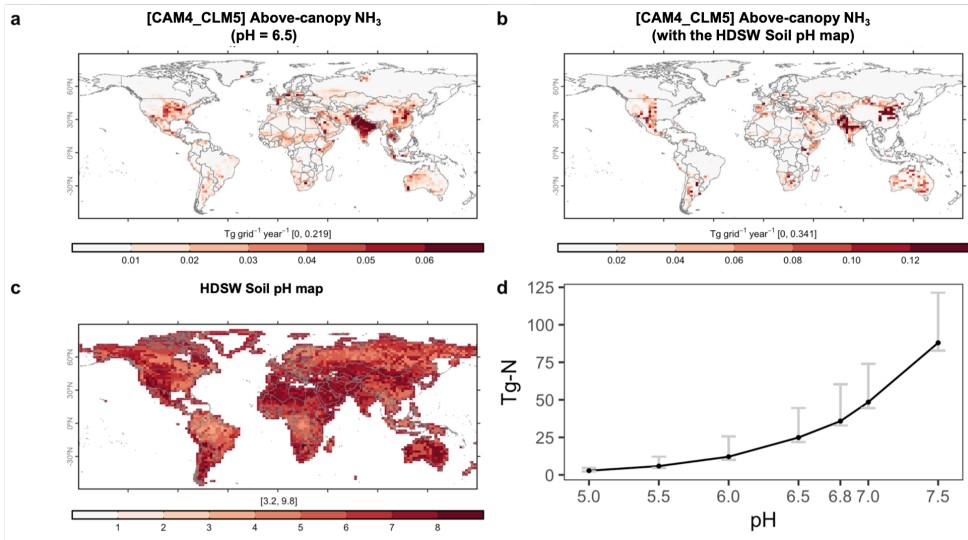


**Figure B1.** CLM5-simulated global soil $NH_3$ emission at above-canopy level with **(a)** a constant global pH of 6.5, and; **(b)** a spatially varying pH map re-gridded from the Harmonized World Soil Database v1.2 (HWSD) (Wieder, 2014), which is shown in map **(c)**. Graph **(d)** shows the total soil $NH_3$ emission against various global pH values from 5.0 to 7.5. Error bars indicate the maxima and minima across five years of simulation.


## Code availability

The modified codes of CESM2 developed in this study will be available when this manuscript is accepted.

## Author Contribution

All co-authors participated in designing the experiments. KMF and MVM developed the model code. KMF performed the simulations. KFM prepared the manuscript with contributions from all co-authors.



## Competing Interests

The authors declare that they have no conflict of interest.

## Acknowledgments

This work was supported by the Research Grants Council (RGC) General Research Fund (Project #: 14323116) awarded to A. P. K. Tai. M. Val Martin acknowledges funding from the Leverhulme Trust through a Leverhulme Research Centre Award (RC-2015-029). We would also like to acknowledge the high-performance computing support from Cheyenne (doi:10.5065/D6RX99HX) provided by NCAR's Computational and Information Systems Laboratory, sponsored by the National Science Foundation.

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
