# Peer review of "Modeling the interinfluence of fertilizer-induced NH3 emission, nitrogen deposition, and aerosol radiative effects using modified CESM2"

_Biogeosciences, 2021_

## Author Response (AR1)

**Responses to the Editor**

Comment:

*Please also carefully check the level of N fertiliser application rate used in the simulations. From the top of my head, the number given 68 TgN/yr is low biased compared to other estimtates.*

Response:

We thank the editor's comments. Our new estimates are that annual fertilization is 96.5 Tg-N/yr in the present day (2000) and 117 Tg-N/yr in our future scenarios (2050). These numbers are more comparable to the values suggested by FAO than the old estimate of 68 Tg-N/yr (when manure was excluded).

We have edited the text to reflect this update. In 3rd paragraph of the revised manuscript:

"The total fertilization rate at the 2050 level was 117 Tg-N yr$^{-1}$ (or +21% from the present-day total fertilization rate at 96.5 Tg-N yr$^{-1}$, **which is comparable to ~100 Tg-N yr$^{-1}$ suggested by FAO (2008)**)."

Reference:
Food and Agriculture Organization of the United Nations: Current world fertilizer trends and outlook to 2011/12, 2008.

**Responses to Reviewer 1**

Comment:

*This paper explores some of the interactions and feedbacks of a fully coupled nitrogen cycle by examining the interactive coupling between nitrogen emissions from synthetic fertilizer, the resulting nitrogen deposition, the impact on climate, terrestrial ecosystems and crops. The paper first introduces a parameterization for nitrogen emissions from synthetic fertilizer, then evaluates it, and then finally examines various feedbacks with interactive nitrogen emissions.*

*While the topic is interesting, the paper requires substantial improvement in all aspects prior to publication as detailed below.*

Response:

We thank the reviewer for their comments and suggestions to improve the manuscript. We have revised the manuscript to address the reviewer's feedback as below. The revised manuscript (with subsequent changes highlighted) and the updated **Supplementary Information** are also attached in the Supplement files.

Comment:

*1. The explanation of the scheme for nitrogen emissions (section 2.2) needs to be more complete.*

*While there is a general reference to the DNDC model at the beginning of the derivation of NH3 emissions from synthetic fertilizer, it is somewhat of a mystery where the specific equations come from. Please explicitly include a rationale for the formulation of the specific equations, especially equations 1, 2 and 7.*

Response:

**Eq. 1** – **Eq. 7** are borrowed directly from DNDCv9.5 (Li et al., 2012; Gilhespy et al., 2014; source code of DNDC v9.5 provided by Changsheng Li via personal communication on Jun 18th, 2015, "the source code" hereinafter). Specifically, **Eq. 1** is from Li et al. (2012); **Eq. 2** is from Li et al. (1992) and Dutta et al. (2016); **Eq. 7** is based on the source code (Li et al., 1992; Gardner, 1965).

We have extended the description of the NH3 scheme in **Section 2.2** as suggested by the reviewer.

Comment:

*Equation (1) is in terms of $f_{vol}$, the fraction of the non-adsorbed aqueous NH3 that volatilizes into NH3 gas. This does not seem to include an explicit term for the partitioning between NH3(aq) and NH3(g). If not, why not? Other formulations include this term.*

Response:

**Eq. 1** provides information to calculate the concentration of gaseous and aqueous $NH_3$, i.e. the partitioning between $NH_{3(aq)}$ and $NH_{3\ (g)}$: the non-adsorbed $[NH_4^+{}_{(aq)}]$ is given by $[NH_4^+{}_{(soil)}](1 - f_{ads})$, $[NH_{3(aq)}]$ by $[NH_4^+{}_{(soil)}](1 - f_{abs})f_{dis}$, and $[NH_{3(g)}]$ by $[NH_4^+{}_{(soil)}](1 - f_{ads})f_{dis}f_{vol}$.

Comment:

*Where does the formulation for $f_{vol}$ come from (equation 7)? This equation seems to have the peculiar property that for a one layer model (l_max=l) there are no emissions, while for a very thin layer this fraction will be maximum.*

Response:

**Eq. 7** is obtained directly from the source code of DNDC v9.5. The variables *l* and *l*_max refer to the depth of a particular soil layer and the maximum depth of a soil column, but not their thickness. In other words, the last term in **Eq. 7** of a shallower soil layer is larger than that of a deeper layer, reflecting the fact that $NH_3$ gaseous from a shallower layer has a higher tendency to be emitted to the surface than that from a deeper layer.

We have made this discussion clearer in the description of **Eq. 7** in **Section 2.2**.

Comment:

*Are the emissions sensitive to the vertical depth profile of fertilizer application? If yes, what is the depth profile of application?*

Response:

Yes, emissions are sensitive to the vertical soil profile. Details of the soil profile structure is tabulated below, which is now added as **Table S1** in the revised Supplementary Information. Soil $NH_4^+$ pool exists in the first 20 layers, which are all prone to volatilization based on our scheme. Fertilizer N and depositional N are added to each soil layer based on a N Input Distribution Fraction, which is a fraction contribution of the predefined weighting factor, $e^{(-10l)/\Delta l}$, where $l$ is the depth of a soil layer and $\Delta l$ the layer thickness, both in meters.

**Table A1. Soil layer structure.**

| Layer# | Layer Node Depth $l$ (m) | Layer Thickness $\Delta l$ (m) | Weighting Factor $e^{(-10l)/\Delta l}$ | N Input Distribution Fraction |
|---|---|---|---|---|
| 1 | 0.01 | 0.02 | 45.2 | 62.8% |
| 2 | 0.04 | 0.04 | 16.8 | 23.3% |
| 3 | 0.09 | 0.06 | 6.8 | 9.4% |
| 4 | 0.16 | 0.08 | 2.5 | 3.5% |
| 5 | 0.26 | 0.12 | 0.6 | 0.9% |
| 6 | 0.40 | 0.16 | 0.1 | 0.2% |
| 7 | 0.58 | 0.20 | 0.0 | - |
| 8 | 0.80 | 0.24 | 0.0 | - |
| 9 | 1.06 | 0.28 | 0.0 | - |
| 10 | 1.36 | 0.32 | 0.0 | - |
| 11 | 1.70 | 0.36 | 0.0 | - |
| 12 | 2.08 | 0.40 | 0.0 | - |
| 13 | 2.50 | 0.44 | 0.0 | - |
| 14 | 2.99 | 0.54 | 0.0 | - |
| 15 | 3.58 | 0.64 | 0.0 | - |
| 16 | 4.27 | 0.74 | 0.0 | - |
| 17 | 5.06 | 0.84 | 0.0 | - |
| 18 | 5.95 | 0.94 | 0.0 | - |
| 19 | 6.94 | 1.04 | 0.0 | - |
| 20 | 8.03 | 1.14 | 0.0 | - |
| 21 | 9.80 | 2.39 | 0.0 | - |
| 22 | 13.33 | 4.68 | 0.0 | - |
| 23 | 19.48 | 7.64 | 0.0 | - |
| 24 | 28.87 | 11.14 | 0.0 | - |
| 25 | 42.00 | 15.12 | 0.0 | - |

Reference:

- Section 2.2.2 in CLM5 technical notes, https://escomp.github.io/ctsm-docs/versions/release-clm5.0/html/tech_note/Ecosystem/CLM50_Tech_Note_Ecosystem.html, accessed on Jul 29, 2021
- Source code of CLM5: https://github.com/ESCOMP/CTSM/blob/master/src/soilbiogeochem/SoilBiogeochemVerticalProfileMod.F90, accessed on Jul 29, 2021

Comment:

*Most formulations use the resistance approach for emissions from the surface which seems appropriate. The formulation presented here appears not to include resistances either for emissions into the canopy or from the canopy to the atmosphere. What is the evidence that this type of approach is valid? It also appears that NH3 from any soil layer is emitted directly into the atmosphere. What is the justification or rationale for this?*

Response:

In our scheme, the term $f_{vol}$ encapsulate wind speed, temperature of soil, and soil layer thickness to quantify the tendency of soil $NH_{3(aq)}$ to break the soil-air interface and vaporize to $NH_{3(g)}$ at the ground surface. The variables considered in this approach are similar to the boundary-layer resistance in several resistance methods (e.g., Pleim et al., 2013).

Our scheme assumes that there is vertical diffusion of $NH_3(g)$ from a deeper soil layer to the surface, but does not explicitly simulate it. Instead, this is represented in the last term in **Eq. 7** as a ratio of $(l_{max} - l)/l$ for the $NH_{3(g)}$ contained in each soil layer. As a result, $NH_4^+$ in a deeper layer is also subject to loss to $NH_3$ volatilization, but at much slower rate than the upper layers. We have added this information in the revised description of **Eq. 7** in **Section 2.2**.

Comment:

*A number of models have incorporated a bidirectional flux of ammonia emissions. The results from the model presented here are indeed more complex than some of the simplest schemes used, but they seem somewhat less complex than the bidirectional approach in Pleim et al (2019) or Zhu et al (2015). Comparisons to these other more complex schemes should also be made in the text and these additional papers should be referenced.*

Response:

We acknowledge that there are limitations in the $NH_3$ volatilization scheme in DNDC v9.5. We decided to implement this scheme as it is one of the models of intermediate complexity that have been developed and validated in multiple studies against field observations. The process-based nature of this scheme can also allow us to evaluate the response of $NH_3$ emission to soil climate, soil nitrogen content, fertilization, deposition, competition against other soil biogeochemical processes (nitrification, microbial uptake, etc.), and vegetation growth. Comparing to other approaches, the DNDC scheme requires variables that are mostly already modeled in CLM5, allowing us to largely capture the dynamic nature of $NH_3$ emission.

We added the citations in the revised manuscript in the last paragraph of **Introduction** and compared our approach to other more complex schemes mentioned by the reviewer.

Comment:

*2. I find the results from the model are rather minimally evaluated and in some cases the evaluation is questionable. This is a bit strange as in lines 72-86 the paper outlines various measurement techniques for evaluation of NH3 emissions, but these are not used in the paper. This seems a little incongruous, as certainly the paper could have used more detailed model-measurement analysis, particularly the N deposition. Better evaluation is needed. While I could understand a minimalistic evaluation if the section on the feedbacks in the system were presented with more depth (see comments below) this is not the case.*

Response:

We thank the reviewer for this note and have revised the introduction to focus on more relevant content.

We decided to use IASI satellite observations and the emission inventories because these datasets provide a global coverage, which allow us to widely compare the results of our global simulations in a consistent manner, including regions with few observations, e.g., South America and Africa. The emission inventories also better match the spatial resolution of our model simulations.

As pointed out by the reviewer, a key result in this study is to evaluate the sensitivity of $NH_3$ and grain production to the dynamic N cycle under intensified fertilization. Hence, we decided to focus our analysis on the benchmarking exercise to qualitatively compare the model performance with our scheme against observations and emission inventories, but not to exhaustively evaluate or improve the model-observation mismatch. We have also extended our analysis and discussion on the feedbacks in the system, as done in **Section 3.3**.

Comment:

*-The model evaluation is in part against other established inventories EDGAR, CMIP and MASAGE (although arguably not with state of the art inventories such as the HTAP.v2.2 inventory and the CEDS inventory, which include significant local information into their emission estimates). The trouble is the EDGAR and CMIP inventories do not separate out manure and synthetic fertilizer emissions from agricultural soils. The paper assumes 1/3 of these emissions are "fertilizer associated" (line 364) where I assume the authors mean synthetic fertilizers. This number may be roughly valid globally, but certainly not regionally valid. Regionally, there may be very different apportionments between manure and fertilizer emissions from soils. Consequently, the geographic comparisons and statistical analysis between the CAM4_CLM5 EDGAR and CMIP6 are likely subject to significant local errors and are therefore not suitable for a quantitative comparison of the inventories. The temporal comparison between these emission inventories is also suspect as NH3 emissions of manure from agricultural soils may have a different seasonality than those from synthetic fertilizer.*

Response:

We agree with the reviewer that excluding manure from our analysis make our model-emission inventory comparison a bit inconsistent. We have re-run our simulations to include manure and modified the manuscript substantially accordingly for the results presented in **Section 3**. The addition of manure did not change the main results of this paper, but simply improved the model-inventory comparison. We would like to note however that model-inventory comparisons are not exact given that our runs are performed using free-running dynamics and thus do not match with the meteorological year of the inventories, while the synthetic fertilizer use is not identical to the ones assumed when inventories were compiled. Thus, these results are presented as qualitative comparisons to indicate where our estimation is consistent with the inventories and where it is not. We also added this clarification to the revised manuscript in **Section 2.4**.

We also note that our named CMIP6 emission inventory is actually the CEDS emission inventory and made that clearer in the revised manuscript.

Comment:

*Two model simulations are compared against the IASI satellite measurements: CAM4_CLM5 and CAM4_CMIP6. It is not clear to me (although maybe I missed it) if the CAM4_CLM5 and the CAM4_CMIP6 emissions are identical except for the synthetic fertilizer NH3 emissions. Are the other sources of NH3 emissions identical so that the only difference in these simulations is from differences in the synthetic fertilizer emissions? Differences in the simulations can only be attributed to the simulated ammonia emissions if the inventories are identical except for the emissions from agricultural soils. Not only do the NH3 emissions from other sectors besides synthetic fertilizer need to be the same between the simulations, but also the NOx and sulfate emissions as the partitioning of NH3 into the aerosol phase depends sensitively on these emissions also. Thus, unless the emission inventories are identical except for the ammonia emissions from fertilizer, it seems difficult, without more analysis, to quantitatively compare CAM4_CLM5 and CAM4_CMIP6.*

Response:

The emission inventories are identical, except for the difference in the fertilize-induced $NH_3$ estimated by CLM5. We have made this point clearer in **Section 3.2** stating that:

"Source of non-fertilizer related $NH_3$ and other reactive gases were identical in these two cases."

*3. The model simulations are not well explained.*

*A number of simulations were made with various feedbacks enabled. However, it is unclear how long any of these simulations were run for, whether ensemble simulations were made and the statistical significance of any difference between the simulations. In these coupled simulations meteorological variability can result in apparent differences. The regional changes in temperature (S6, for example), are quite large, probably larger than can be expected from rather small changes in radiative forcing.*

Response:

We took the advice of the reviewer and decided to extend the duration of our simulations.

We agree with the reviewer that meteorological variability might affect our coupled simulations as only 5-year averages were used originally. We therefore decided to extend the duration of our simulations to 30 years to minimize that influence. We detailed our new set up as well as made clearer our simulation description in Section 2.3:

"All simulations were run for 30 years using the spun-up year-2000 initial conditions with the corresponding land cover data provided out-of-the-box by CLM5. The first 10 years of outputs were used to further stabilize the model (such that the change in annual emission fluxes < ±10%) after our ammonia scheme was implemented. Our analysis in the next section focuses on the averages of last 20 years of simulated results to minimize influence from any long-term meteorological variability."

We also included results of two-sample t-tests to highlight where differences are statistically significant in our figures.

Comment:

*Biogeochemical models of soils are notoriously difficult to deal with, as they are notoriously difficult to spin up to equilibrium. In these simulations this is not discussed. Please elaborate on the spinup of the biogeochemical part of these simulations including the extent to which the coupled system was spun up to equilibrium. What state was the model initialized from? How did this state change with the introduction of the new parameterization? Was the model spun up to equilibrium after making changes to the parameterization of ammonia emissions?*

Response:

We used the out-of-the-box spun-up initial condition provided by CLM5. We identified the annual-total ammonia emission fluxes fluctuate mildly after the first few years of simulations (<10% yr$^{-1}$). Hence, we set to use the first 10 years to stabilize the simulations for our modifications.

We provided this information in the revised manuscript, **Section 2.3**, as cited in our responses to the comment above.

Comment:

*There are some feedbacks with the crop model which need to be addressed. It appears that only synthetic fertilizer is added to the crops whereas in reality manure is also be used to fertilize crops. Consequently, the crops are likely significantly under-fertilized in the model simulations. The apparent under fertilization in these simulations would suggest that the crops take up a greater fraction of added nitrogen in the model-world than they would in reality. This would suggest the fraction of applied fertilizer volatilized is underestimated in the model. (In reality surplus nitrogen is added to agricultural systems, a fraction of which is lost.) Please address the extent to which this might impact the simulations.*

Response:

In the revised manuscript, we have included manure (at $+2$ g-N m$^{-2}$ s$^{-1}$ in addition to synthetic fertilizer as in default CLM5) in our simulations. Our present-day fully-coupled case [CAM4_CLM5_2000] estimates that 68 Tg-N yr$^{-1}$ of synthetic fertilizer is applied annually, and it is increased to 96 Tg-N yr (+41%) when manure is included. The corresponding grain-N produced increases from 18.5 Tg-N yr$^{-1}$ to 22.1 Tg-N yr$^{-1}$ (+19%). As suggested by reviewer 2, in this study, we have also determined the nitrogen-use efficient (NUE) as the ratio of grain-N produced to fertilizer input, and the nitrogen leakage ratio (NLR) as the ratio of NH$_3$ emission rate to fertilization rate. The NUE is reduced from 27% (synthetic fertilizer only) to 23% (synthetic and manure fertilizer), indicating that the crops are not under-fertilized.

The ammonia emission is correspondingly increased from 10.5 Tg-N yr$^{-1}$ to 14.4 Tg-N yr$^{-1}$ (+38%). We also find that NLR is slightly reduced from 15.1% to 15.0%, implying that NH$_3$ volatilization is likely not underestimated in the cases with and without manure fertilizers.

Details are provided in **Table 3**, **Section 3.1**.

Comment:

*It seems likely that with the under-fertilization the simulated crops may not show sufficient growth. This is likely to have climate impacts including changes in latent and sensible heat flux and changes in albedo. How does crop growth in the scheme documented here compare with that in the standard CLM model without the ammonia emissions? How does this impact the radiative budget?*

Response:

There are some differences in crop growth, which is partly manifested in the change in crop yield/production. Without our $NH_3$ scheme and the dynamic land-atmosphere nitrogen cycle, the standard model estimates 3.2% more annual total grain production than the fully coupled case. This difference is expected as $NH_3$ volatilization is a substantial competitor to soil ammonium and implementing our $NH_3$ scheme would reduce soil $NH_4^+$ available for plant uptake.

Spatially, larger differences are seen in the northern US and Europe as shown in the figure below:

Change in Grain Production (Tg-(dry matter) $yr^{-1}$)

[Standard Model] – [CAM4_CLM5_2000]

[Figure]

Regarding the radiative budget, we find the following results and added to Section 3.3:

"Compared to the default model, our fully coupled simulation estimated a 0.13 W m$^{-2}$ increase in global downward radiative flux, which is substantial compared to the total aerosol radiative forcing of +1.0 Wm$^{-2}$ (Myhre et al., 2013)."

Comment:

*4. I found the section regarding model sensitivities needs significant more in depth analysis. It seems to me this section could be the real novelty of the paper (schemes with bidirectional fluxes have been implemented previously as have prognostic equations for NH3 emissions). This is particularly true as the model evaluation is not comprehensive.*

*In section 3.3 the different responses of the system are simulated after a change in forcing (i.e., a change in added fertilizer).  If I understand correctly the authors are comparing the [CAM4_CLM5] with 2000-level fertilization with: (i) [CAM4_CLM5] with a 30% increase in fertilization, with (ii) CAM4_CLM5_CLIM with a 30% increase in fertilization but constant nitrogen deposition and with (iii) CAM4_CLM5_NDEP with a 30% increase in fertilization but constant aerosol forcing. I found the section somewhat confusing, perhaps in part due to the notation. It would probably be clearer if the authors distinguished in their notation the simulations with different emissions (e.g., the CAM4_CLM5 with standard emissions versus that with a 30% increase in fertilization).*

Response:

We appreciate the reviewer's suggestion and have updated the notations of our simulations with different fertilization levels with corresponding suffixes, i.e., "_2000: and "_2050". As mentioned above, we also extended substantially the discussion in **Section 3.3** to strengthen the results from our model sensitivity analysis.

Comment:

*Also, how was the aerosol forcing kept constant in CAM4_CLM5_NDEP?*

Response:

We chose a configuration in CAM4-chem that makes the atmospheric chemistry inactive to the radiative transfer module. We have made this point clearer in the manuscript, **Section 3.3**:

"Similarly, [CAM4_CLM5_NDEP_2050] was set up such that addition/reduction of $NH_3$-induced aerosols would be inactive to the radiative transfer module, i.e., would not induce changes in aerosol-climate interactions, so that we could isolate the impact of $NH_y$ deposition on $NH_3$ emission and crop growth."

Comment:

*In CAM4_CLM5 emissions increase by 27% or to 2.4 Tg N/year. In CAM4_CLM5_CLIM with constant year 2000 deposition fluxes the emissions increase to 2.5 Tg N/year; in CAM4_CLM5_NDEP and constant aerosol forcing they increase to 2.7 Tg N/year. It is hard to interpret the significance of these differences as they seem small on the face of it. Are these differences really significant? What is the difference in radiative forcing? Substantial more analysis could be conducted here. As just one example the paper states some changes are "likely a consequence of better vegetation growth driven by increased NHy deposition following higher NH3 emissions". This can be evaluated by examining the model.*

Response:

The reviewer raises an important point with respect to the significance of the results. We have conducted a series of two-sample t-tests when comparing variables and added indications for statistically significant results in all relevant figures as suggested. For example, in Supplementary Information, we provide **Figure S11** to illustrate the changes in annual-mean net downward radiation flux at the Earth's surface:

[Figure]

To examine vegetation growth, we used crop grain production as the increase in grain biomass during the grain filling period is a direct measure of crop growth as well as the nitrogen use efficient for crop food products. As in CLM, crop grain production is one key indicator of crop growth (Levis et al., 2012) and farming efficiency. We decided to focus our results on the reported grain production.

Comment:

*I am rather puzzled by the statement (Lines 601, 602): "We estimated that the effect of nitrogen deposition on NH3 emission is +2.7 Tg-N yr–1 globally" with a reference to Figure 5. Figure 5 shows the changes in emissions when fertilizer is increased by 30% compared to the case with no change in emissions. Shouldn't the 2.7 Tg-N yr–1 increase in emissions in CAM4-CLM-NDEP be, to a large extent, attributable to the increase in fertilizer, not to the effect of nitrogen deposition. Maybe I have completely missed something here.*

Response:

The statement was referring to the change in $NH_3$ from [CAM4_CLM5] with present-day level fertilizer to [CAM_CLM5_NDEP] with fertilizer at the future level. We have clarified our statement and rewrote that sentence. It reads now as:

"We also estimated that if the synthetic fertilizer use was to increase by 30% from 2000's level, $NH_3$ emission would rise by 3.3 Tg-N $yr^{-1}$ globally (see **Figure 5**)."

Comment:

*Other points:*

*-Lines 104-105: "Recent inventories…..", but then the paper quotes Sutton (2013). There are in fact much more recent inventories than that. Other more recent inventories include the HTAP_v2.2 inventory and the CEDS inventory which are not mentioned.*

Response:

We welcome the reviewer's suggestion to include results from more recent inventories and have revised it accordingly. Our CMIP6 emission inventory is actually the CEDS inventory, and we have updated the manuscript accordingly to clarify this information:

"CAM4-chem employs a bulk aerosol approach and predicts the formation of $PM_{2.5}$ components including $SO_4^{2-}$, $NO_3^-$, and $NH_4^+$, where the injection rates of precursors – sulfur dioxide ($SO_2$), $NO_x$, and $NH_3$ – are prescribed by Coupled Model Intercomparison Project phase 6 (CMIP6)/Community Emissions Data System (CEDS) emission inventory for anthropogenic activities as well as biomass burning in the default configuration (Hoesly et al., 2018)."

Comment:

*-Lines 274- 295 Emissions of other reactive nitrogen compounds. As far as I can the emissions of species other than NH3 are not discussed in the paper or evaluated. This section can then be omitted. It seems to me what is pertinent here is the loss of ammonia through nitrification.*

Response:

We moved this section to **Supplementary Information** to improve the readability.

Comment:

*- Line 212: I assume that equation (1) should also include other loss terms: washout and nitrification for example. Please clarify.*

The potential soil $NH_3$ emission rate determined by **Eq. (1)** is used by the model to compute the competition for available soil $NH_4^+$ with other processes, namely, plant uptake, microbial immobilization, and nitrification. We expanded the sentence to describe this treatment more explicitly. The model assumes leaching (including "washout") occurs for soil nitrate only.

We added the addition information to the 2nd and the 5th paragraphs of **Section 2.2** in the revised manuscript.

Comment:

*-Lines 325-327: In equation (9) why is Vc set to a fixed deposition velocity instead of the deposition used in the chemistry model?*

Response:

We wanted to be as consistent as possible with the DNDC configuration and kept the $V_c$ for in-canopy $NH_3$ constant as provided in the DNDC scheme (Li et al., 1992; Nõmmik, 1965).

Comment:

*- The constants in a number of the equations in section 2.2 do not have defined units (e.g., equations 4 and 5). Please give explicitly where these equations come from as appropriate and the units for the constants.*

Response:

We added the references and units of constants to the equations in **Section 2.2** as suggested.

Comment:

*-The change in ammonia is apparently calculated for each soil layer (equation 1), but I assume that equation (8) is in terms of all soil layers. Please clarify.*

Response:

In **Eq. (8)**, $f_{can}$ is a column-level variable. It is applied to the column-total actual $NH_3$ emission flux (g-N m$^{-2}$ s$^{-1}$). We clarified this point as suggested.

Comment:

*-Line 305: "In our coupled simulations, we omitted the portion of NH3 emission associated with synthetic fertilizer from the inventory input for CAM4-chem."It is not clear where the inventory for this input comes from in CAM4-chem.*

Response:

We revised the sentence to emphasize that the portion of fertilizer-induced NH3 is omitted from the CMIP6/CEDS emission inventory: (**Section 2.3**)

"In our coupled simulations, we substituted the portion of $NH_3$ emission associated with synthetic fertilizer from the CAM4-chem inventory input (CESD) for our online simulated emission rates from CLM5. Atmospheric $NH_3$ and $NH_4^+$ formed sequentially return to CLM5 through deposition."

Comment:

*-Line 409: Some synthetic fertilizers have a much smaller ammonia volatilization loss than urea.*

Response:

We agree with the reviewer's comment. MASAGE includes fertilizers that can be more or less prone to $NH_3$ volatilization than urea. We revised the sentence to reflect such view:

"For example, MASAGE considers multiple-type fertilizers that can be more or less prone to $NH_3$ loss than urea (Bouwman et al., 2002), and assumes a three-stage fertilization at sowing, growth, and harvesting (Paulot et al., 2014)."

Comment:

*-Figure 5. Please include figure captions for the figure components: a, b, c and d. Also the caption on Figure 5b is misleading. It would be helpful if the two cases were distinguished.*

Response:

Captions of **Figure 5** and **Figure 6** are edited according to the reviewer's suggestions.

**Responses to Reviewer 2**

Comment:

*This paper presents a parameterization for evaluating ammonia (NH3) volatilization from soils within the Commonity Eath System Model (CESM). The authors couple both the emission and deposition fluxes between the land and atmospheric components of the CESM, and use this capability to evaluate the emission of NH3 following fertilizer applications and the transport, deposition and possible re-emission of the emitted NH3. They perform further simulations to evaluate the effect of increased fertilizer usage on modeled ammonia emissions and crop harvests and the role of atmospheric feedbacks in these responses.*

*The two-way land-atmosphere exchange of ammonia has been simulated in a number of other models, although not within CESM. However, introducing the bi-directional NH3 exchange into the nitrogen cycle of an Earth system model does open up new possibilities for assessing the role of atmospheric transport of NH3 in the global nitrogen cascade. The coupled simulations in this manuscript present an interesting step towards this direction.*

*Thus, I find the manuscript in principle suitable for publication Biogeosciences. However, I also have several concerns related to the model formulation and the experiments, as detailed below. Addressing these will probably require major revisions before the paper can be published.*

Response:

We thank the reviewer for their comments and suggestions to improve the manuscript. We have revised the manuscript to address the reviewer's feedback as below. The revised manuscript (with subsequent changes highlighted) and the updated **Supplementary Information** are also attached in the Supplement files.

Comment:

*1. Model formulation and evaluation*

*The model equations (1-9) are supposed to be derived from the DNDC model. However, except for the chemical equilibria in Eqs (3)-(6), which are fairly standard, I haven't found the equations in the DNDC literature cited. It is possible that I have missed something, since there exist multiple versions of DNDC. But if the DNDC is only a source of inspiration, then the authors should not write that their model is "derived from" the DNDC. In the current form, the model equations look like plausible but rather ad-hoc parameterizations for the volatilization process.*

Response:

As suggested also by reviewer 1 we have improved the description of the NH3 volatilization scheme.

**Eq. (1)** – **(8)** are directly implemented from DNDC, corresponding citations are added to the manuscripts.

**Eq. (7)** is obtained from the source code of DNDCv9.5 shared by Changsheng Li with us via personal communication on Jun 18[th], 2015.

**Eq. (9)** in our paper is from the Eq. (17) from the DNDC v9.5 Scientific Basis and Processes (Institute for the Study of Earth, Oceans, and Space, University of New Hampshire, 2017), with additional terms of $b(h_{top} - h_{bot})$ to account for the height effect of the plant canopies, referring to Eq. (11) in (Pleim et al., 2013) for calculating the in-canopy aerodynamic resistance.

Comment:

*Most of the existing models for NH3 volatilization and exchange have been verified with at least some field data. Here, the model is evaluated by comparing with existing inventories and by comparing the simulated NH3 columns with the IASI data on yearly level. This makes sense, but as the only source of empirical evaluation, it has the issue that the geographic variation predicted by the model becomes conflated with the variation in the N input. In other words, some of the resolved variation is likely to originate in the geographic distribution of fertilizer use. Only a fraction of the global NH3 emission is from synthetic fertilizers, which further weakens the signal.*

*I understand that there is no easy way around this, but it would be good to see how the model predicts the geographic distribution of the ratio between NH3-N emitted and fertilizer-N applied.*

Response:

We welcome the reviewer's suggestion and have calculated two ratios, nitrogen leakage ratio (NLR) = $NH_3$ emission / fertilizer N input, and nitrogen use efficiency = grain N harvested / fertilizer N input, to evaluate the loss and conversion of fertilizer N to $NH_3$ and grain production. We have added this discussion in **Section 3.3** and **Table 4:**

"**Table 4** summarizes the changes in annual-total fertilizer-induced $NH_3$ emission estimated by these simulations when the global synthetic fertilizer use rises to 130% of the 2000 level. The total fertilization rate at the 2050 level was 117 Tg-N $yr^{-1}$ (or +21% from the present-day total fertilization rate, which is comparable to ~100 Tg-N $yr^{-1}$ suggested by FAO (2008)). We also computed the nitrogen leakage ratio (NLR) and nitrogen use efficiency (NUE) for each case. NLR maintains at ~15% for [CAM4_CLM5_2000] and [CAM4_CLM5_2050] while NUE decreases from 23% to 22%, respectively, indicating that the crops are under nitrogen surplus under this future fertilization scenario. This is also confirmed by the reduced ratio of crop uptake to fertilization from ~130% to ~115% (**Table S3**).

We also note that the DNDC $NH_3$ emission algorithm itself has been well validated with field observations, as explained in the first paragraph of **Section 2.2** in the revised manuscript.

Comment:

*Also, for easier comparison with existing inventories, it would be useful to provide the regional emission totals.*

Response:

Regional emission totals are used to compute the statistics in **Figure 2**. The information is now included in the manuscript as **Table 3**.

Comment:

*General comments*

*Eq. (1): How do you define the potential emission rate? The left hand side is a time derivative of a NH3 concentration (?), which is generally not the same as the NH3 flux to the surface. The factor 1/(delta t) on the right hand side does not make sense: the flux per time unit cannot depend on the timestep of the model.*

Response:

As mentioned above, we have revised the notations of the variables in the equations to clarify the physical meaning of each term in the $NH_3$ volatilization scheme.

Comment:

*How does the deposited NH4+ enter the soil pools? Is 100 % of the wet deposition assumed to enter soil, or do you consider surface runoff? Figure 1 indicates that NOx deposition goes to the NH4+ pool, is this really the case?*

Response:

In CLM5 (and in our parameterization), the depositional nitrogen from dry and wet deposition of nitrate and $NH_4^+$ both enter the soil $NH_4^+$ pools (Lawrence et al., 2019). Loss of soil nitrogen via runoff/leaching only happen to nitrate in the model.

Comment:

*Eq. (2): What is the source of this formula? Does it mean that f_ads is greater than 1 when f_clay is very small? Why?*

Response:

**Eq (2)** is from the source code of DNDCv9.5 (Li et al., 1992; Nõmmik, 1965). We implemented upper and lower bounds to f_abs such that its value is within 0 and 1 so $f_{ads}$ will be close to 1 but will not be larger than unity when $f_{clay}$ is small. We added the information regarding the upper and lower limit of $f_{ads}$ to the revised manuscript.

*Eq. (7): Please give a rationale for this equation. What is the role of T_soil here, given that it already appears in Eqs. (4) and (5)? Why does the flux from a given layer depend on the thickness of the soil column below? If you evaluate the emission from deeper layers, shouldn't there be also exchange between the layers? Finally, why parameterize the exchange between the soil and the atmosphere using the wind speed instead of using the resistance formulation already present in CLM for calculating dry deposition?*

Response:

According to DNDCv9.5, **Eq. (7)** encapsulates the effects of soil temperature, wind speed, and depth on gas diffusion along the depth of the soil. While DNDC does not directly compute the gas exchange between layers, the last term in **Eq. (7)** provides an estimation of how much gaseous $NH_3$ from a layer at depth $l$ will reach the surface air immediately above the soil column, i.e., quantifying the tendency of $NH_3$ vapor in the soil to break the interface to enter the atmosphere. $T_{soil}$ in Eq. (4) and Eq. (5) is used to determine the reaction rates that govern $NH_4^+/NH_3$ equilibrium in the soil.

We adopt the parameterization in DNDC for the soil-atmosphere exchange instead of the CLM dry deposition formulation so that our scheme was as more consistent as possible with DNDC. It is now explained in greater detail in the 4th paragraph in **Section 2.2**.

Comment:

*How does the evaluation of the volatilization flux relate to the other NH4-consuming processes? Does it contribute to the N demand similar to the plants and microbial immobilization?*

Response:

The model distributes available soil $NH_4^+$ to all competing processes according to their relative demands (individual potential flux to sum of all four potential fluxes) without bias toward any process (Lawrence et al., 2019). The processes are: nitrification (flux size = ~500 Tg-N yr$^{-1}$), immobilization (~1600 Tg-N yr$^{-1}$), plant uptake (~900 Tg-N yr$^{-1}$) and $NH_3$ volatilization (14 Tg N/yr). Thus, when $NH_3$ emission is introduced, the 14 Tg-N yr$^{-1}$ is taken partially from plant uptake and partially from microbial immobilization. These are now explained in the 5th paragraph in **Section 2.2**.

Comment:

*Eqs (8) and (9): Which equations in the DNDC document do you refer to? Also, I'm not sure of what Eq. (8) means – it gives a linear relation between the concentration in soil and in the canopy, but what about the flux? What is the rationale for the 1/s factor? What if the wind is calm? Finally, the constant 14 m-1 seems to go back to the the paper of Erisman et al. (1994), where it is given as an empirical constant in a certain resistance component. How does it correct for the effect of canopy thickness?*

Response:

As mentioned above, we have revised the notations of the variables in the equations to clarify the physical meaning of each term for the $NH_3$ volatilization scheme in **Section 2.2**.

**Eq. (8)** ($F_{atm}$) is the portion of soil $NH_3$ emission flux that is not capture by the canopy and is released to the atmosphere. We followed the DNDC scheme and discussed in **Section 2.2** that "dividing soil $NH_3$ emission rate by $s_{10}$ gives an approximate in-canopy $NH_3$ concentration". Hence, when the wind speed is low, the in-canopy $NH_3$ concentration will be higher (i.e., slower dispersion) under a constant soil $NH_3$ emission rate.

Our scheme takes into account the effect of canopy thickness in **Eq. (9)**, which is based on the Eq. (17) from the DNDC v9.5 Scientific Basis and Processes (Institute for the Study of Earth, Oceans, and Space, University of New Hampshire, 2017). The additional terms $b(h_{top} - h_{bot})$ account for the height effect of the plant canopies, i.e., canopy thickness. We used Eq. (11) in Pleim et al., (2013) for calculating the in-canopy aerodynamic resistance.

Comment:

*2. Experiment setup*

*More details are needed about the model setup.*

*First, describe how the aerosol-radiation interaction was evaluated. Direct and/or indirect effects, nitrates, sulfates?*

Response:

As indicated by the reviewer, we extended the model setup description and added the following information to the manuscript:

"Atmospheric $NH_y$ does not directly interact with radiative transfer in CAM4-chem. Instead, its radiative implications are manifested in the radiative effect of changes in sulfate formation (direct) and the sequential sulfate-induced changes in cloud optical properties (indirect). Detailed description of the radiative transfer processes in CAM4-chem is provided in Lamarque et al. (2012) and the model manual (CAM Reference Manual, 2021).

Comment:

*Second, please describe whether the runs used some kind of nudging of the meteorological fields. This would be very important for understanding the comparisons between the runs which are presented later.*

Response:

The simulations were run with free dynamics. We revised the manuscript to provide more information about this in **Section 2.3**:

"CAM4-chem was run with free dynamics in the standard spatial resolution of 1.9º by 2.5º horizontally with 27 vertical layers (from surface to ~40 km). CLM5 was run in the same horizontal resolution with 25 soil layers down to ~50 m below ground. Sea surface temperature (SST) and sea ice conditions (Hurrell et al., 2008), as well as the mixing ratios of greenhouse gases (Meinshausen et al., 2017) were all fixed at the 2000-levels."

Comment:

*Are CAM and CLM the only active components in the simulation?*

Response:

We clarified this point in **Section 2.3** as:

"Only the atmosphere (CAM4-chem) and the land (CLM5) components were active."

Comment:

*If the model is driven or nudged by atmospheric reanalysis data (I think this is called the "offline" configuration in Lamarque et al. (2012)), the different simulations will share the same meteorological variability. However, if the simulations are run with fully prognostic atmospheric dynamics, the simulations will develop chaotic variations, and in this case, five years is unlikely to be long enough to obtain statistically significant differences. If the results shown are indeed from a free-running CAM simulation, all comparisons of means between the configurations should be tested for statistical significance to rule out the effect of the internal variability. The large differences in parameters like surface temperature over remote areas (Fig. S6) suggest that this might be an issue.*

Response:

Our simulations were run under "free dynamics" and not "offline" nor "specific dynamics". To address the concern of long-term interannual meteorological variability, we extended our simulation to 30 years, as also suggested by reviewer 1. We provided details in **Section 2.3** in the revised manuscript:

"All simulations were run for 30 years using the spun-up year-2000 initial conditions with the corresponding land cover data provided out-of-the-box by CLM5. The first 10 years of outputs were used to further stabilize the model (such that the change in annual emission fluxes $< \pm 10\%$) after our ammonia scheme was implemented. Our analysis in the next section focuses on the averages of last 20 years of simulated results to minimize influence from any long-term meteorological variability."

We also conducted a series of two-sample t-tests to determine where the meteorological changes are significant and included the results in corresponding figures.

Comment:

*The experiments using modified fertilization rates (Section 3.3) should be introduced in the methods (Section 2.4). It might be worth noting that the future increases in agricultural production might involve also expansion of agricultural land area, and thus, the fertilizer application rate might on some areas change differently from the total fertilizer use. This would affect the response of nonlinear processes.*

Response:

We welcome the reviewer's point. We have now focused our analysis on the feedback effect of increased fertilizer use, as extended now in the last paragraph of **Section 2.3**, including a note to highlight the potential influence of cropland expansion in our discussion:

"We note that future increases in agricultural production might also involve cropland expansion, but such practice is not included in this study."

Comment:

*3. Results*

*Regardless of the statistical aspects, some of the findings seem non-trivial and should be backed up with more analysis.*

*First, fairly large differences between the configurations are attributed to aerosol radiative effects. Please show the differences in the aerosol load (e.g. AOD) and in the aerosol radiative forcing, perhaps split by aerosol type if relevant. Are you able to rule out other atmospheric feedbacks, for example due to changed evapotranspiration?*

Response:

We appreciate the reviewer's suggestion and have added the contrast of $NH_4^+$ and sulfate burden, net downward radiation fluxes of our simulations in **Figure S11** to **Figure S13** in the **Supplementary Information** and discussed the details in the last paragraph in **Section 3.3** of the revised manuscript.

Comment:

*Second, the results for grain production under increased N fertilization seem surprising. Generally increasing N fertilization would be expected increase harvest yield, even if not linearly. Here, the effect is negative for many regions, especially those in the southern hemisphere. What causes this? The authors should verify that the fertilization response in the CLM crop model is realistic (perhaps on a regional or per-crop basis) because otherwise the discussion of atmospheric feedbacks on crop production is not very meaningful.*

Response:

Lombardozzi et al. (2020) has extensively studied the crop response to fertilization in CLM5. Though increased N input would likely raise grain yield (Lombardozzi et al., 2020), a warmer temperature can also shorten crop growth period as the grain is harvested immediately when crop growing-degree-day (GDD) reach the maturity threshold, which may shorten the grain-filling period and result in smaller grain mass at harvest (Levis et al., 2012). We see that effect in our results as, for example, grain production decreases in South America (**Figure 8b)** coinciding with warmer surface temperatures (**Figure S9b)**. We have now extended our discussion of these results in the second last paragraph in **Section 3.3**.

Comment:

*Finally, I'm a bit surprised to see such a big difference in crop growth (Fig. 6) between CAM4_CLM and CAM4_CLM_CLIM (i.e. due to deposition) over areas like China or the U.S. corn belt, where N fertilization rates are known to be high. How large is the difference in the annual N deposition flux, and how does it compare to the annual N fertilization per crop area?*

Response:

The annual total global N deposition rate ranges from 15 to 16 Tg-N yr$^{-1}$, which is ~16% relative to the model "present-day" fertilization rate (96.5 Tg-N yr$^{-1}$) or ~13% relative to the "future" rate (117 Tg-N yr$^{-1}$). The difference in grain production associated with the increased N deposition is revealed in **Figure 8** and **Figure S8** of the revised manuscript. The figures show that the increased N deposition are substantial, specially over China and US Corn Belt, and likely enhanced grain yield.

Comment:

*Specific comments*

*Introduction: the intro is not bad, but could be shortened to give a stronger focus on the present work. For example, the paragraph about in-situ observations seems excessive, since none of those data are used here.*

Response:

As indicated by the reviewer as well as reviewer 1, we have revised the Introduction to focus more on the relevant content.

Comment:

*L53: is the spending in USD a global total?*

Response:

Yes, it is referring to a global total. We revised the sentence to reflect this:

"**The global** public health system may have to spend 20–290 billion USD more each year to compensate for the $NH_3$-derived detrimental effects on air quality and health (Gu et al., 2012; Paulot and Jacob, 2014; Guthrie et al., 2018)."

*L142-145: there have been many (non-CESM) modeling studies using the resistance framework to simulate NH3 exchange, including the canopy capture.*

Response:

We thank the reviewer to point out to other non-CESM studies.  We are aware of such studies and revised the manuscript accordingly:

"We also developed a prognostic parameterization for canopy capture of $NH_3$, instead of using a fixed generic value (e.g., one constant canopy reduction factor for all plants as used in some other studies (e.g., Riddick et al., 2016; Bouwman et al., 1997)."

Response:

The suggested citation is added.

Comment:

*Figure 1: Much of the litter N is first assimilated to the microbial biomass and then remains in the soil organic matter (SOM) before becoming mineralized to NH4+. Having a SOM N pool in the figure would make sense, perhaps instead of the microbial N, which is anyway only implicitly represented in CLM (see e.g. https://escomp.github.io/ctsm-docs/versions/master/html/tech_note/Decomposition/CLM50_Tech_Note_Decomposition.html). Also, N2O and NOx are produced by both nitrification and denitrification. Denitrification also produces N2.*

Response:

The microbe N pool is shown to highlight the two processes, namely, N fixation and immobilization associated with the microbes. We renamed the "Litter N" to "Litter/SOM N" to reflect the existence of SOM N in the model. We decided not to show $N_2$ in the diagram as it is an inert species but mentioned it in the figure caption.

Comment:

*Section 2.3: the purpose of this section is unclear, since the rest of the paper is only about NH3.*

Response:

This section has been moved to **Supplementary Information**.

Comment:

*L297: Is this the setup described in Lamarque et al., (2012)?*

Response:

The reviewer is correct: The basic setup is largely similar to Lamarque et al., (2012), which is now cited in the introduction of that section.

Comment:

*L297: Is this the setup described in Lamarque et al., (2012)?*

Comment:

*Section 2.4: do you include any biomass burning emissions of NH3?. If not, aren't you missing part of the N deposition in some regions?*

Response:

Yes, emission of NH3 from biomass burning is prescribed in the CMIP6/CEDS inventory. We supplemented the information in the revised manuscript as:

"CAM4-chem employs a bulk aerosol approach and predicts the formation of $PM_{2.5}$ components including $SO_4^{2-}$, $NO_3^-$, and $NH_4^+$, where the injection rates of precursors – sulfur dioxide ($SO_2$), $NO_x$, and $NH_3$ – are prescribed by Coupled Model Intercomparison Project phase 6 (CMIP6)/Community Emissions Data System (CEDS) emission inventory for anthropogenic activities as well as biomass burning in the default configuration (Hoesly et al., 2018)."

Comment:

*L303: biogenic emissions...of isoprene?*

Response:

Isoprene emission is one of the biogenic emissions handled by MEGAN2.1. We added isoprene as an example in Section 2.3:

"Biogenic emissions, e.g., of isoprene, are updated online from CLM5 using the Model of Emissions of Gases and Aerosols from Nature (MEGAN) version 2.1 (Guenther et al., 2012)."

Comment:

*L315: by boundary layer, do you mean the quasi-laminar layer resistance Rb?*

Response:

Yes, it refers to the laminar sublayer. We hence revised the sentence as:

"For NH3 vapor, the model calculates the aerodynamic and the boundary-layer (laminar sublayer) resistance based on the online atmospheric dynamics, …"

Comment:

*L321: the Henry's law applies to NH3, right?*

Response:

Yes, as stated in the manuscript in Section 2.3:

"For $NH_3$ vapor, the model calculates the aerodynamic and the boundary-layer (laminar sublayer) resistance based on the online atmospheric dynamics, while the surface resistance over land is determined according to the online CLM5 surface variables, e.g., canopy height and LAI, as well as species-specific reactivity factor for oxidation and effective Henry's Law coefficients."

Comment:

*L321: the Henry's law applies to NH3, right?*

Comment:

*L343: Manure N is a significant N source in many areas. What is the reason for omitting it, and how does this affect the model results?*

Response:

The current version of CLM5, manure is assumed to be applied to crop land at a constant rate of 2 g-N $m^{-2}$ $yr^{-1}$ (Lombardozzi et al., 2020), which is ~30% of total fertilizer input. Since the model is yet capable of tracing the source of the soil $NH_4^+$ which then is responsible for the $NH_3$ emission, we originally decided to focus our estimation of the soil $NH_3$ emission that was solely from synthetic fertilizers by omitting the manure fertilizer. In the revised manuscript, we have re-run our simulations to include both synthetic and manure fertilizers and updated our results discussed in **Section 3** in the manuscript substantially accordingly.

Comment:

*L387: Boyland and Russell discuss a certain type of air quality models. I don't think that their conclusion can be used as a universal standard for a quite different application.*

Response:

We agree with the reviewer and revised the sentence to refrain from referring to the "acceptable range".

Comment:

*L410: perhaps even more importantly, some fertilizers have typically much lower NH3 emission factor than urea (e.g. Bouwman et al., 2002). This includes for example anhydrous ammonia, which is common in the US.*

Response:

We agree that MASAGE has included fertilizers which can be more or less prone to $NH_3$ than urea. We revised the sentence in **Section 3.1** to reflect such view:

"MASAGE considers multiple-type fertilizers that can be more or less prone to $NH_3$ loss than urea (Bouwman et al., 2002), and assumes a three-stage fertilization at sowing, growth, and harvesting (Paulot et al., 2014)."

Comment:

*L421: "high spatiotemporal correlation is" unclear, I guess you just mean the temporal correlation. Though, how interesting is it to correlate the model to the inventories, given that the inventories usually prescribe the monthly variation? Why not compare to the IASI data on a monthly basis?*

Response:

We updated the analysis to include monthly IASA data and find that our updated model can reduce model low-biases on a monthly basis compared to the simulation using the CEDS emission inventory. We included this results in **Figure 4** and added relevant discussion in **Section 3.2** of the revised manuscript.

Comment:

*L464: did Hu et al. really use CESM?*

Response:

Thank you for catching this typo. The correct citation shall be He et al., (2015)

He, J., Zhang, Y., Glotfelty, T., He, R., Bennartz, R., Rausch, J., and Sartelet, K.: Decadal simulation and comprehensive evaluation of CESM/CAM5.1 with advanced chemistry, aerosol microphysics, and aerosol-cloud interactions, J. Adv. Model. Earth Syst., 7, 110–141, https://doi.org/10.1002/2014MS000360, 2015.

*Figure 4: the IASI heatmap is saturated over large areas. Can you add more color levels to better show the variability of high-NH3 regions?*

We revised **Figure 4** to address the saturation issue.

Comment:

*L515: "ammonium salts" maybe better "secondary aerosols"*

Response:

We changed "ammonium salts" to "secondary ammonium aerosols".

Comment:

*L515: "ammonium salts" maybe better "secondary aerosols"*

Comment:

*L529: I'm not sure if this kind of nonlinearity is excepted, since the NH3 emission is usually evaluated with constant emission factors. The emission has sometimes been suggested to increase faster than linearly (Jiang et al., 2017). Slower than linear increase seems to imply that either plant or microbial N demand increases nonlinearly to the input. Have you tried to analyze this?*

Response:

We reanalyzed the simulations and have updated the results in **Section 3.3**:

"The super-linear increase in $NH_3$ emission (+24%) relative to total fertilizer (+21%) is associated with sub-linear rise in nitrification (+17%), crop uptake (+5.8%) and other loss processes of soil $NH_4^+$."

Comment:

*L540: I'm not sure if I follow the logic here. What drives such a gradient in plant uptake?*

Response:

We have removed the sentence and updated the results in **Section 3.3**.

Comment:

*L555: is there any further evidence to show that this is a causal connection?*

Response:

The enhanced evapotranspiration due to better vegetation growth tends to shift surface energy balance to latent heat flux from sensible heat flux (Bonan, 2019). This sentence has been removed in the revised manuscript.

Comment:

*L566: is this in Tg of C, dry matter, or something else?*

Response:

Grain production is measured in Tg-(dry matter). Notes on this are added to the same line and the caption of **Figure 6**.

Comment:

*L589: more reliable...than what?*

Response:

Based on our results, we found that the dynamic estimation of $NH_3$ is more reliable than using constant emission inventory values under dynamic climate and environmental conditions. Hence, we revised the sentence as:

"These new features enabled CESM2 to perform, for the first time, a more reliable estimation of soil $NH_3$ emission and atmospheric $NH_3$ concentration than using constant emission inventory values under dynamic climate and environmental conditions."

Comment:

*L610-615: Are the manure management emissions really easier to track? There is a huge diversity in manure management systems around the world. Not all facilities are confined. In many regions, collecting accurate information about farming practices is certainly not a trivial task.*

Response:

We intended to bring up that NH3 emissions from manure management and other sources can be estimated and validated more efficiently using different approaches from ours. We rewrote the part to better convey such message:

"Unlike soil emission whereby the volatilization of $NH_3$ depends on a series of biogeochemical processes, emissions associated with manure management are typically estimated differently, e.g., collecting activity data and emission factors from factory managers, and installing monitoring instruments at outlets of confined facilities, e.g., animal factories (Bouwman et al., 1997; Paulot et al., 2014)."

Comment:

*L620: what data for validation do you have for fertilizer but not manure?*

Response:

For example, we do not have the amounts of manure fertilizers applied for each crop in each region and the $NH_3$ emission attributed to such fertilizers. We have now made this point clearer in the manuscript.

Comment:

*L620: what data for validation do you have for fertilizer but not manure?*

Comment:

*L621: By manure fertilizer, do you mean manure application on crops, or all manure-related sources? But Riddick et al (2016) only considered agricultural emissions, and did not consider different manure management processes.*

Response:

We revised the sentence to reflect the correct information:

"It is noteworthy that manure is attributable up to ~60% of total soil $NH_3$ emission (Vira et al., 2020) and hence shall warrant further research efforts in terms of its downstream impact on ecosystems via nitrogen deposition and aerosol radiative effect."

Comment:

*L629-632: Does the effect of the initial NH4 pools still remain after five years of spinup? What do you mean with a "soil nitrogen map"? I thought that the soil N is evaluated prognostically in CLM.*

Response:

Though the key metric variables in this study, e.g., $NH_3$ emission flux and grain production, have reached a quasi-equilibrium (interannual changes $<\pm10\%$) in a few simulation years, different initial conditions may result in difference steady states of a model simulation. For this, a more realistic "soil nitrogen map" – which we refer to the information regarding the global spatial distribution of soil N content – may help to constrain the modeled N concentration in the soil N pools by providing a more accurate initial conditions for model simulations. We have clarified this in the **Section 4** accordingly:

"The overestimation by CLM5 in this study may point to the more-fertile-than-reality soil conditions in the model, highlighting the need for a more realistic soil nitrogen map compiled by field surveys to better constrain the initial conditions for the model."

*L640: note that the N fertilization rate soybean is usually low, since it is a leguminous crop.*

Response:

Thank you for providing this information. We updated the fertilization rate used in the model in **Section 4**:

"A chamber study suggested that soybean can absorb up to 20 kg-N ha$^{-1}$ of $NH_3$ via leaf capturing (Hutchinson et al., 1972), which is a significant amount compared to average fertilizer use for soybean of 13–45 kg-N ha$^{-1}$ in CLM5"

---

## Author Response (AR2)

**Responses to the Review 1**

Comments:

The authors have rerun the simulations and extended them over several decades, which should indeed be enough to obtain statistically significant differences. The model description is also improved, although the references to the DNDC source code are problematic because the code is not publicly available.

The longer simulations seem to have resolved my biggest concern (statistical significance) in the initial manuscript. There are still some puzzling results which require attention. Unless there is a simple explanation which I'm missing, addressing these issues may still require more revisions.

Reponses:

We again thank the reviewer's constructive feedback. We have revised the manuscript to address the reviewer's comments/suggestions as follows. The revised manuscript (with changes highlighted) is attached.

Comments:

First, it is unclear to me how the runs with aerosol-radiation (apparently also aerosol-cloud) interactions disabled are set up. I understand that in the "interactive" runs, the prognostic aerosol fields are used in the radiative transfer and cloud microphysical calculations. What aerosol distributions are used in the two non-interactive (NDEP) runs? If these are somehow prescribed, then how is it ensured that the differences attributed to the aerosol-climate interactions are really due feedback effects and not just because the prescribed forcing is different from the prognostic aerosol forcing? It would be helpful if the authors would formulate their hypotheses regarding the climate effects more explicitly and then explain how the model experiments test these hypotheses.

Responses:

We thank the reviewer for the opportunity to further explain our experimental design. In our non-interactive runs, the aerosol fields are prescribed. Hence, when comparing with the interactive cases, the differences in radiative budgets would include the effects of both the N deposition, the corresponding feedbacks, and the inherent spatial differences between the prescribed and prognostic aerosols for radiative transfer calculations. We note that the prescribed-prognostic differences for [CAM4_CLM5_NDEP_2050] and [CAM4_CLM5_2050] are substantial for sulfate (up to 4% in zonal mean mass ratio; mostly inland) and dust (up to 30% zonally) and rather negligible for organic carbon, black carbon, and sea salt. Since intense loading of dusts are mostly found in the desert regions, we would be able to characterize the impacts induced by these two kinds of aerosols.

We now provided more details about our configurations and experimental design in the second last paragraph of **Section 2.3:**

"**Table 1** provides configuration details of our experiments. [CAM4_CLM5_2000] and [CAM4_CLM5_2050] encapsulated the full functionality of our implementation, i.e., CAM4-chem receives the online CLM5 $NH_3$ emission rates as input to predict atmospheric $NH_3$ concentration, the subsequent formation of secondary ammonium aerosols (modeled as changes in sulfate aerosols in the model), and the corresponding instantaneous sulfate aerosol radiative effect, whilst CLM5 obtains the online CAM4-chem dry and wet deposition rates of $NH_y$ and $NO_x$ to calculate the addition of soil $NH_4^+$ via deposition. The deposited nitrogen will eventually enrich soil fertility and fuel the re-emission of soil $NH_3$ while the aforementioned aerosol radiative effect can cool the Earth's surface and suppress $NH_3$ volatilization. **The [CAM4_CLM5_NDEP] cases were set to isolate the impact of $NH_y$ deposition on $NH_3$ emission and crop growth. In this setup, CAM4-chem used prescribed gases (except for water vapor) and aerosols in the radiation transfer calculation (i.e., aerosol-radiation interaction is disabled). Hence, any changes in the atmospheric sulfate aerosol loading induced by the addition/reduction of $NH_3$ would not affect radiative transfer. We note that the differences in radiative budget between the [CAM4_CLM5_NDEP] and other configurations with aerosol-radiation interaction enabled would include the effects attributable to both $NH_3$-induced sulfate changes as well as the differences in spatial distribution between the prescribed and prognostic aerosols. For instance, the differences**

**between [CAM4_CLM5_NDEP] and [CAM4_CLM5] are substantial for sulfate (up to 4% in zonal mean mass ratio; mostly inland) and dust (up to 30% zonally; mostly in Sub-Saharan Africa and other desert regions), and are unlikely related to NH$_3$ changes; meanwhile, the differences are rather negligible for organic carbon, black carbon, and sea salt. This configuration was intended to isolate the enhanced fertilization effect of N deposition. Similarly,** [CAM4_CLM5_CLIM] cases were prescribed with constant nitrogen deposition fluxes so that we could quantify the impacts of the changes in instantaneous aerosol radiative effects. **We hypothesized that an increased NH$_3$ emission would promote the formation of sulfate aerosols, and the subsequent aerosol cooling effect would be observed in this setup. Finally,** we further evaluated the impacts of intensive fertilizer use to promote agricultural production in the future as projected by FAO (2007) by repeating the first three simulations with fertilization at present-day (2000; model default) and future (2050; assuming 30% more synthetic fertilizers while manure fertilizer is kept at 2000-level) rates. We note that future increases in agricultural production might also involve cropland expansion, but such practice was not included in this study."

Comments:

Second, the radiative forcing due to NH3 emissions has been thought to be small, negative and mainly due to nitrates (see e.g., IPCC AR5 or AR6). But the differences between the CAM4_CLM5_2050 and CAM4_CLM5_NDEP_2050 (runs with and without aerosol feedbacks) in Figs. S9 and S11 are large, with about 1 K change in surface temperature and and down to 20 W m-2 difference in net surface radiation over large areas. Even more surprisingly, the differences occur over areas like the Sub-Saharan Africa where the simulated NH3 emissions seem quite small. The areas affected partly coincide with a large change in the sulfate concentration, but the increase of sulfates over Africa seems to coincide with a higher downward radiation flux and higher surface temperature. Wouldn't we expect a cooling effect from the sulfates? What is the mechanism here? I don't remember ever seeing such a big effect attributed emissions of ammonia.

Responses:

We would like to clarify that

1) **Figure S11** is illustrating the local annual mean of the radiative forcing at the surface level for the N deposition and for the aerosol-radiation interaction, both measured by local changes in net (SW + LW) downward radiation at the model surface level, over pixels with croplands. These metrics are not directly comparable to the radiative forcing reported in, e.g., IPCC AR5 or AR6, which is usually the difference in area-weighted global mean (direct) radiative effects of aerosols (measured as net downward SW + LW radiation) at the top of atmosphere, under present-day and preindustrial emissions.

   For comparison, the global (over all land types and oceans) annual mean radiative forcing induced by the increased fertilization, i.e., [CAM4_CLM5_2050] vs [CAM4_CLM5_2000], is −0.005 W m$^{-2}$.

2) The large differences in downward radiation over the Sub-Saharan Africa in **Figure S11(b)** are largely due to the differences between the prescribed and prognostic dusts, rather than the reduced emission of NH$_3$. The more negative values (greener) in **Figure S11(b)** indeed indicate stronger cooling effects, where coincide in the lower surface temperature (greener) in **Figure S9(b)**. The cooler condition over the Sub-Saharan Africa from 3) has resulted in reduced NH$_3$ emission (greener) in **Figure 7(b)** as well as less sulfate aerosols (greener) in **Figure S12(b)**.

   To make these points clearer in the manuscript, we added these descriptions and results as follows:

   In the last paragraph of **Section 3.3**:
   "**Comparing the 2000- and 2050-fertilization levels, our fully coupled simulation estimated a –0.005 W m$^{-2}$** of global downward radiative flux **(i.e., cooling),** which is

**virtually negligible compared to the 16-model mean total anthropogenic aerosol radiative forcing of –0.27 W m$^{-2}$ reported in Myhre et al. (2013).**"

In the last paragraph of **Section 3.3**:

"Though the global impact is also negligible (+0.004 W m$^{-2}$), $\Delta_{clim}$ reveals a substantial regional cooling at the surface level (**Figure S11(b)**) largely because there are more prognostic dusts in [CAM4_CLM5_2050] over Sub-Saharan Africa than prescribed in [CAM4_CLM5_NDEP_2050]. Such cooling effect results in lower surface temperature (**Figure S9(b)**) and also suppressing the formation of particulate sulfate (**Figure S12(b)**)."

We also provided more details in the second last paragraph of **Section 2.3**, which helps clarify what the differences in sulfate and dust between the simulations actually entail. See our revisions and responses above.

Comments:

The differences caused by the nitrogen deposition (CAM4_CLM5_2050 vs CAM4_CLM5_CLIM_2050) seem smaller, as expected. But even here, some of the regional differences seem large. For example, in Fig. 6, CAM4_CLM5_CLIM_2050 shows only 2 % increase in grain production for Europe, while CAM4_CLM5_2050 with interactive N deposition has a 17 % increase, as if the effect of deposition was greater than the fertilization itself. Is this difference realistic and statistically robust?

Responses:

The changes concerned were checked and found to be insignificant, and were removed from our discussion; these did not affect the major conclusions of our paper.

Comments:

Specific comments (line numbers refer to the track changes version)

Eq. 1: It's still unclear how F(soil, pot) is converted to a flux (per unit of time). If Eq. (1) gives the maximum flux per timestep, then what happens if the length of timestep is changed?

Responses:

This equation was originally developed in DNDC to estimate the potential amount of soil $NH_3$ to be emitted per day. In this study, we converted the unit of this daily value from per day to per second, so that we can get the total amount of $NH_3$ emitted in each time step (30 min in this study). Then, this daily value is recalculated in the next time step. A longer timestep would reduce the frequency of such recalculation.

Comments:

Eq. 2 is actually OK, I misread it on the first go.

Responses:

Thank you for your feedback.

Comments:

Eq. 7: Is lmax 42 m (per Table S1)? We know that even relatively shallow (5-10 cm) incorporation of fertilizers reduces the NH3 emissions by 50 % or more, so it could be expected that the depth-dependent factor would be close to zero for all but the top 5-10 layers or so.

Responses:

Yes, the reviewer is correct, $l_{max}$ corresponds to the bottom layer of soil. $NH_3$ emission from the deeper layers contributes very little to the total column emission.

Comments:

L318: Hoesly et al. (2018) state that "CEDS data do not include open burning, e.g., forest and grassland fires, and agricultural waste burning on fields, which was developed by van Marle et al. (2017)." These are important sources of NH3. Please check.

Responses:

We have revised the sentence to clarify that biomass burning in our model are prescribed using the emission inventories described by von Marle et al. The updated sentence now reads:

"CAM4-chem employs a bulk aerosol approach and predicts the formation of $PM_{2.5}$ components including $SO_4^{2-}$, $NO_3^-$, and $NH_4^+$, where the injection rates of precursors – sulfur dioxide ($SO_2$), $NO_x$, and $NH_3$ – are prescribed by the Coupled Model Intercomparison Project phase 6 (CMIP6)/Community Emissions Data System (CEDS) emission inventory (CMIP6 hereinafter) for anthropogenic activities (Hoesly et al., 2018). **The biomass burning emissions used for our simulations are described by von Marle et al. (2016, 2017) and are all assumed as surface emissions without plume-rise nor predefined vertical distribution.**"

Comments:

L346: I actually do not find a detailed description of the radiative transfer in CAM4 in Lamarque et al. (2012) or in Collins et al. (2006). However, Lamarque et al. states that no cloud-aerosol interaction is available in CAM4, which appears to contradict with what is stated here. A better description of the simulated aerosol-radiation and aerosol-cloud effects is needed, since they are critical for some of the model experiments.

Responses:

We thank the reviewer for noting that our description of the CAM4 radiate transfer was not fully correct. We updated the description in the 3$^{rd}$ paragraph in **Section 2.3** and provided a more appropriate reference:

"**In the default configuration, atmospheric chemistry interacts with the climate solely through radiation in CAM4-chem (Lamarque et al., 2012). Furthermore, atmospheric reactive nitrogen (NH$_4^+$ or NO$_3^-$) does not directly interact with radiative transfer in the model. Instead, its radiative implications are manifested via altering the gas-aqueous partitioning of sulfate (Emmons et al., 2010; Metzger, 2002) and the subsequent changes in direct radiative effect due to any changes in sulfate aerosols. The subsequent sulfate-induced changes in cloud optical properties (indirect radiative effect) were not considered in this work. Detailed description of the radiative transfer processes in CAM4-chem is provided in Neale et al. (2010).**"

Neale et al (2010) Description of the NCAR Community Atmosphere Model (CAM 4.0), https://www.cesm.ucar.edu/models/ccsm4.0/cam/docs/description/cam4_desc.pdf

We also corrected the description in the last paragraph of **Section 3.3**:

"On the other hand, we expected the **sulfate aerosols** induced by agricultural NH$_3$, which directly increases aerosol albedo , would reduce the amount of insolation reaching the Earth's surface."

L618: So why is it warmer in the 2050 fertilization scenario?

Responses:

After considering the reviewer's comments and reexamining the results, we found that the Asian warmer surface temperature and grain production increases are not uniformly statistically significant and decided that temperature may not be the important factor. We hence removed the concerning discussion from the main text.

Comments:

L647: The overall aerosol radiative forcing is thought to be negative. It seems that the best estimate for total aerosol effect in AR5 (p. 662) was -0.9 W m-2.

Responses:

We thank the reviewer for rising this point. As indicated above we added the following discussion:

"Comparing the 2000- and 2050-fertilization levels, our fully coupled simulation estimated a $-0.005$ W m$^{-2}$ in global downward radiative flux (i.e., cooling), which is virtually neglectable compared to **the 16-model mean total anthropogenic aerosol radiative forcing of $-0.27$ W m$^{-2}$ reported in Myhre et al. (2013)**."

**Responses to Reviewer 2**

Comments:

I found the revised paper improved over the previous version, but still have some concerns. The introduction reads quite well and except for a few minor points looks good. The model description also well. The description of the model simulations and results were much clearer.

All line numbers refer to the version with author tracked comments.

Responses:

We thank the reviewer for their constructive feedback. We have revised the manuscript to address the reviewer's comments/suggestions as below. The revised manuscript (with changes highlighted) is attached.

Comments:

Major Comments
1. The authors should carefully distinguish between fertilizer (referring to synthetic fertilizer and manure) and synthetic fertilizer. They should clearly state the difference. In some places it was rather confusing as to what the authors were referring. Instances are referred to below. This should be easily remedied.

It appears that Fig. 2 is probably only from synthetic fertilizers, yet the caption says "fertilizer induced" which would imply manure also? The numbers from MESAGE look like they are from synthetic fertilizer, but it is not clear how one obtains synthetic fertilizer emissions from EDGAR and CMIP6. To my knowledge those latter inventories give agricultural soil manure emissions which include manure, fertilizer and grazing emissions. To what extent does the added manure in the present manuscript represent the grazing component?

Responses:

As suggested by the reviewer, we have clarified throughout the manuscript what type of fertilizer our study refers to.

In **Section 3.1**, 1st paragraph:
"We extracted the monthly fertilizer-induced $NH_3$ emission estimates from MASAGE, and assumed that one-third of the total agricultural $NH_3$ emission reported by CMIP6 and EDGAR are **associated with synthetic fertilizer**, which is consistent with the apportionment reported in previous studies and environmental reports (Paulot et al., 2014; Riddick et al., 2016; National Oceanic and Atmospheric Administration, 2000; European Environment Agency, 2010; Gu et al., 2012; Paulot et al., 2015; Zheng et al., 2017)."

In **Section 3.1**, 3rd paragraph:
"Our estimate is higher than all three inventories of $NH_3$ emissions associated with synthetic fertilizers, which are 10 Tg-N yr$^{-1}$ for CMIP6 and EDGAR, and 9.1 Tg-N yr$^{-1}$ for MASAGE."

Caption of **Figure 2**:
"Fertilizer-induced $NH_3$ emission estimated by CLM5 **(synthetic and manure)** and other emission inventories **(synthetic only)**."

We also wanted to note that $NH_3$ emission due to grazing is not considered as the emission associated with manure applications in this study, and clarified this in the text too:.

In **Section 2.3**, 1st paragraph:
"…for our online simulated emission rates from CLM5. **This study did not consider manure spreading on pastures and grazing animals.** Atmospheric $NH_3$ and $NH_4^+$ formed sequentially return…"

2. Using a constant manure fertilization rate is somewhat surprising as there are global distributions of manure application (e.g., Zhang et al., 2017). The rate incorporated (2 g-N s–1) was presumably used in the CLM so crops would grow even in regions where synthetic fertilizer was lacking. This rate does not seem appropriate to represent the regional distribution of manure fertilizer for emissions. Presumably using a constant rate implies that crops in regions largely fertilized largely with synthetic fertilizer the added nitrogen is much higher than reality? Thus the detailed comparison between the various inventories and CAM4_CLM5_2000 is difficult to interpret. Is this due to the emission scheme or due to a simplistic estimate of applied manure? You might compare the manure N rates used here to those from an established inventory. The units here don't make sense for added manure don't makes sense: 2 g-N s–1.

Responses:

Thank you for pointing this out, and we have corrected the unit as g-N $m^{-2}$ $yr^{-1}$. Now the relevant sentence in **Section 2.1**, 2nd paragraph, reads:
"Manure fertilizer application rate is assumed constant for all crops at 2 **g-N $m^{-2}$ $yr^{-1}$**, same as the model default (Lombardozzi et al., 2020)."

This application rate is a default value of CLM5 (Lombardozzi et al., 2020). Though we agree a more realistic development of manure application would be a big improvement, expanding the capacity of CLM5 to include spatial and temporal varying manure application rates is beyond the scope of this study.

Comments:

3. It is difficult to understand how the authors set up the sensitivity simulations. It is very important to clearly define the simulation one is assessing the sensitivity against so as to interpret the results.

For the deposition rates the paper states: "The deposition rates were prescribed in the default configuration and dynamically computed by CAM4-chem in our version." (l199-l201). It appears then in both CAM4_CLM5_CLIM_2000 and CAM4_CLM5_CLIM_2050 the deposition rates are the same and taken from the CAM4_CLM5 assuming 2000- level fertilization (Table 1), where I take it that CAM4_CLM5 has fixed deposition. Since the lifetime of NH3 and NH4+ in the atmosphere is short the deposition should be equal to the emissions on an annual basis. So by comparing the depositions in these model runs the authors are essentially comparing the emissions in the interactive model (CAM4_CLM5_CLIM) and the implied emissions in CAM4_CLM5. This comparison does not seem to get at the importance of interactive emissions, but at the difference between the different emissions, which would not be expected to be the same. A quantification of the effect of interactive emissions in the CAM4_CLM5_CLIM simulations have on the emissions themselves (through deposition) seems difficult. At any rate from the explanations in the paper I don't see how it was done.

Likewise for aerosols it appears that the "NH3-induced aerosols would be inactive" (l578). This seems to imply that this aspect is just examining the importance of a simulation with NH3-induced aerosols to one without. The differences, then, between these simulations have little to do with interactive emissions, but with including ammonia emissions at all. This could have been done, just as well, with fixed emission inventories. Moreover, it is unclear what is assumed with regard to sulfate aerosols – I assume it is only the ammonium nitrate aerosols that are impacted.

Responses:

To make the explanation of our experimental design clearer, we expanded the description of our modeling experimental design in Table 1. For that, we moved an introductory paragraph from **Section 3.3** to **Section 2.3** to explain our experimental designs earlier in the paper. We also rewrote the paragraph to provide more design details, as also suggested by Reviewer 1.

The second last paragraph in **Section 2.3** now reads:

[revised manuscript text omitted]

Comments:

-l97-99 "near real time high-resolution maps of atmospheric NH3". The maps shown in Van Damme et al., 2018 are time averaged over a rather long periods. Are near real time maps of satellite NH3 really accurate?

Reponses:

We agree with the reviewer's comment and have deleted "near real-time". The revised sentence now reads:

"It enables the **creation of high-resolution maps** of atmospheric $NH_3$ and the possibility of pinpointing industrial and agricultural emission hotspots with diameters smaller than 50 km (Van Damme et al., 2018)."

Comments:

-l108 "agricultural emissions". It would be good to define these precisely. As used does this term include emissions from both manure management (mostly from barns and storage) and from agricultural soils (which includes grazing animals, manure spreading onto pastures and other cropland and synthetic fertilizer emissions)?

Reponses:

We expanded the sentence to provide more information regarding the "agricultural emissions" here.

"Recent inventories adjusted the estimated agricultural emissions (**including manure management, and both synthetic and manure fertilizers**) in 2000–2010 to 33–37 Tg-N yr$^{-1}$ (Sutton et al., 2013; Janssens-Maenhout et al., 2015; Hoesly et al., 2018)."

Comments:

-l139-l144 Vira et al., 2020 and Vira et al. 2021 also used the CESM.

Responses:

We added the two citations as suggested:

"Many studies have employed CESM for studying processes in both the atmospheric and terrestrial nitrogen cycles, e.g., $NO_x$ and $N_2O$ emission (Saikawa et al., 2013, 2014; Zhao et al., 2017), deposition (Lamarque et al., 2013), denitrification and nitrate leaching (Nevison et al., 2016), crop nitrogen uptake (Levis et al., 2018), and reactive nitrogen input to ecosystem associated with synthetic and manure fertilizers (Riddick et al., 2016; **Vira et al., 2020, 2021**)"

Comments:

-l153 'DNDC' is just kind of thrown in there. Please give the abbreviation, plus maybe a short introduction to the model

Responses:

We agree with the reviewer's suggestion and have edited the sentence as:

"Comparing to other approaches, **our scheme, which borrowed from a standalone biogeochemical model, the DeNitrification-DeComposition (DNDC),** requires variables that are mostly already modeled in CLM5, allowing us to largely capture the dynamic nature of $NH_3$ emission."

Comments:

-l200 All deposition N is added as NH4+. Is there a reference for this? Lawrence et al. (2020) does not seem to document this (although maybe I missed it).

Responses:

The model treatment of N deposition is explained in **Section 22.3** of Lawrence *et al.* (2018) Technical Description of version 5.0 of the Community Land Model (CLM) (https://www.cesm.ucar.edu/models/cesm2/land/CLM50_Tech_Note.pdf).

We added this citation in our manuscript:

"All added depositional and fertilizers N are added to the soil $NH_4^+$ pool of each layer from ground surface to 0.4 m underground according to a model-defined soil profile (**Table S1**) (Lawrence et al., 2018)."

Comments:

-Were interactive soil NOx emissions included in these simulations? Or were they fixed.

Responses:

Interactive soil $NO_x$ emissions were not included as inputs for the atmospheric model in our simulations. All $NO_x$ emissions for CAM4-chem were from emission inventories.

We added a sentence to the 1st paragraph in **Section 2.3** to clarify this:

"**We note that the $NO_x$ emission inputs for CAM4-chem were solely from the emission inventories and did not include those from our modified denitrification and nitrification schemes.**"

Comments:

-l258 " evaluated the uncertainty" to pH. Please provide a brief summary of the findings in the main part of the paper

Responses:

We added a summary there in the revise manuscript to improve clarity as suggested by the reviewer:

"**Briefly, a higher pH would promote model $NH_3$ emission rate exponentially as the emission rate is of the order of $10^{pH}$. This high sensitivity warrants the need to include crucial chemical processes in the model for accurately determining soil pH online.**"

Comments:

-l260-265 this explanation should probably be included after equation (7), not before it.

Responses:

We agree with the reviewer and have rearranged the paragraph as:

"Lastly, we used this equation to calculate $f_{vol}$ (Li et al., 1992; Gardner, 1965; source code of DNDC v9.5):

$$f_{vol} = \left(\frac{1.5s}{1\,[\text{m s}^{-1}]+s}\right)\left(\frac{T_{soil}}{50\,[\degree\text{C}]+T_{soil}}\right)\left(\frac{l_{max}-l}{l_{max}}\right) \quad (1)$$

where $s$ (m s$^{-1}$) is surface wind speed; $T_{soil}$ (°C) is soil temperature; $l$ and $l_{max}$ (both in m) are the depth of each soil layer and the maximum depth of a soil column, respectively. **Our scheme assumes that vaporized soil NH$_3$ in a deeper layer diffuse upward to the surface, but does not explicitly simulates the process. Instead, it is represented in the last term in Eq. (7) as a ratio of $(l_{max} - l)/l$ for the NH$_{3(g)}$ contained in each soil layer. Hence, soil NH$_4^+$ in deeper layers is also subject to loss to NH$_3$ volatilization but at much slower rates than that in the upper layers.** Details of the soil profile are provided in **Table S1**. The actual soil NH$_3$ to be emitted ($F_{soil,act}$; g-N m$^{-3}$) from each soil layer is then determined by the lower of the $F_{soil,pot}$ or the available soil NH$_4^+$ after competition with other processes, namely, plant uptake, microbial immobilization, and nitrification. The model distributes available soil NH$_4^+$ to all competing processes according to their relative demands (individual potential flux to sum of all four potential fluxes) without bias toward any process (Lawrence et al., 2019). The column-level actual soil NH$_3$ emission flux ($F_{soil}$; g-N m$^{-2}$ s$^{-1}$) is computed as the sum of the product of the $F_{soil,act}$ and layer thickness (m) at each layer, and assumed to emit to the atmosphere constantly over a model time step size ($\Delta t = 1800$ s in this study)."

Comments:

-l301-303 "dividing soil NH3 emission rate by s10 gives an approximate in-canopy NH3 concentration, and multiplying the latter with vc and L produces an estimated quantity of NH3 retained by the canopy". This is not obvious to me – is there a reference or can you clarify?

Responses:

We appreciate the opportunity to explain this.

Consider the in-canopy volume as a rectangular box with a base area of 1 m by 1 m. Assume also that there is a constant inflow of $NH_3$ at $F$ g-N $m^{-2}$ $s^{-1}$ from below into the box. On the other hand, some wind (at a speed $s_{10}$ m $s^{-1}$ from an arbitrary direction) replaces the air inside the box with some fresh air (assuming no or very little $NH_3$). Dividing $F$ by $s_{10}$ would give us an equilibrium concentration of $NH_3$, say $C$ g-N $m^{-3}$, that still remains in the box. This $C$ g-N $m^{-3}$ of $NH_3$ is what the canopy would be exposed to. Multiplying this $C$ g-N $m^{-3}$ of $NH_3$ with the depositional velocity ($v_c$ m $s^{-1}$) and the LAI ($L$ $m^2$ $m^{-2}$) would give us the $NH_3$ being captured/retained by the canopy (in g-N $m^{-2}$ $s^{-1}$).

Comments:

-l322 "we substituted the portion of NH3 emission associated with synthetic fertilizer". This is confusing. Why just synthetic fertilizer when you are considering both synthetic fertilizer and manure. And according to Hoesly et al. (2018) sectors in CEDS includes Agricultural Manure-management and Soil-emissions. I don't see how you can subtract out synthetic fertilizer. Moreover, soil emissions usually include manure spreading on pastures and grazing animals. The latter two are not included in the author's estimate, correct? Additional clarification is necessary here.

Responses:

Thank you for catching this. We meant to say fertilizer, and not synthetic fertilizer.

Regarding the CEDS, we made an assumption as explained in the first paragraph of **Section 3.1**:

"We … assumed that one-third of the total agricultural $NH_3$ emission reported by CMIP6 and EDGAR are associated with synthetic fertilizer, which is consistent with the apportionment reported in previous studies and environmental reports (Paulot et al., 2014; Riddick et al., 2016; National Oceanic and Atmospheric Administration, 2000; European Environment Agency, 2010; Gu et al., 2012; Paulot et al., 2015; Zheng et al., 2017)."

The reviewer is correct that this study did not consider manure spreading on pastures and grazing animals.

Hence, we clarified these points in the paragraph and it now reads: :

"In our coupled simulations, we substituted the portion of $NH_3$ emission associated with **fertilizers** from the CAM4-chem inventory input (CESD) for our online simulated emission rates from CLM5. Atmospheric $NH_3$ and $NH_4^+$ formed sequentially return to CLM5 through deposition. **This study did not consider manure spreading on pastures and grazing animals.**"

Comments:

-l342, are the effects of atmospheric reactive nitrogen on ozone or methane included?

Reponses:

We did not access the changes in ozone or methane, as it is beyond the scope of our study. Our speculation is that, as $NH_3$ reacts with atmospheric $NO_2$, changes in the spatial distribution in $NO_2$ associated with the new $NH_3$ scheme might affect the ozone chemistry and likely also methane lifetime accordingly.

Comments:

-l358 "change in annual emission fluxes <=10%" - does this mean the long-term trend is less than 10% or the interannual variability. A long term trend of 10% seems rather significant.

Reponses:

Here the 10% refer to interannual variability of the emission fluxes. We have revised the sentence to clarify this:

"The first 10 years of outputs were used to further stabilize the model (such that **the interannual variability of the emission fluxes could be** < ±10%) after our ammonia scheme was implemented."

Comments:

-l401 "one-third are fertilizer-associated". First, please clarify what is meant by fertilizer here: synthetic fertilizer or synthetic and manure fertilizers. It might be helpful to explain the emission sectors from the inventories.

Responses:

We agree with the reviewer's suggestion and have updated the sentence to reflect that it's associated with synthetic fertilizers:

"We extracted the monthly fertilizer-induced $NH_3$ emission estimates from MASAGE, and assumed that one-third of the total agricultural $NH_3$ emission reported by CMIP6 and EDGAR are **associated with synthetic fertilizer**, which is consistent with the apportionment reported in previous studies and environmental reports (Paulot et al., 2014; Riddick et al., 2016; National Oceanic and Atmospheric Administration, 2000; European Environment Agency, 2010; Gu et al., 2012; Paulot et al., 2015; Zheng et al., 2017)."

Comments:

-Table 3 should clearly state where the numbers are estimated and where they are straight from the inventory. It should be possible to put some error bounds on these numbers from the literature.

Responses:

We edited the caption of **Table 3** to clarify that the simulated $NH_3$ are estimated in this study while other values are reported by the inventories. The caption now reads:

"Regional fertilizer-induced $NH_3$ emission totals estimated by our model and reported by other inventories."

As state in the manuscript that these model-inventory comparisons are not meant to be exact but shall be considered as qualitative ones. We decided to include 1-SD's for the multi-year estimations / inventory values while maintain how other values are presented in **Table 3:**.

**Table 3.** Regional fertilizer-induced $NH_3$ emission totals (Tg-N $yr^{-1}$) estimated by our model and reported by other inventories.

| | Global | USA | Europe | India | China |
|---|---|---|---|---|---|
| **CAM4_CLM5_2000**[a] | 14.2 (± 0.60) | 2.1 (± 0.35) | 0.6 (± 0.07) | 2.8 (± 0.28) | 1.2 (± 0.08) |
| **CMIP6**[b] | 10.9 (± 0.65) | 0.9 (± 0.05) | 0.7 (± 0.16) | 1.9 (± 0.15) | 2.6 (± 0.30) |
| **EDGAR**[c] | 10.5 | 0.7 | 0.5 | 1.6 | 3.0 |
| **MASAGE** | 9.1 | 0.5 | 0.4 | 1.7 | 2.8 |

a  20-year 1-SD are shown in the brackets
b  16-year 1-SD are shown in the brackets (2000–2015)
c  Variation in 2012 of both cultural statistics and emission factors ranged from 186% to 294% (Crippa et al., 2018)

Comments:

-l421 Vira et al (2020) gives the contribution from synthetic fertilizer which I think you are comparing against.

Responses:

We made our comparison with Vira et al (2020) clearer and the text reads now:

"Our estimation is close to the 12 Tg-N yr$^{-1}$ (from synthetic fertilizer only) and 18 Tg-N yr$^{-1}$ **(11 Tg-N yr$^{-1}$ from both synthetic fertilizer and 6.5 Tg-N yr$^{-1}$ from manure application)** reported by two similar studies, Riddick et al. (2016) and Vira et al. (2020), respectively."

Comments:

-Since ammonia emissions from manure dominate over those from synthetic fertilizer, and the inventories give emissions from agricultural soils why aren't the combined manure, synthetic fertilizer emissions being compared.

Responses:

The manure $NH_3$ reported in the emission inventories includes both from manure fertilizer and management, but accurately disaggregating the reported emission rates is not straightforward and out of the scope of this study. Such comparison would also require further simulations with exact meteorology inputs matching with the inventories. We hence decided to present the qualitative comparison in this study, as stated in **Section 2.4** of the manuscript.

Comments:

-l480-483. This is confusing. EDGAR contains manure management.

Responses:

We meant here that the source data of EDGAR we used in this study accounted for $NH_3$ emission from both synthetic & manure fertilizers, while CMIP6's $NH_3$ emission included also manure management.

We have revised the sentence to clarify this:

"EDGAR and CMIP6 have higher background levels than MASAGE because the original estimates **used in this study** accounted for not only synthetic fertilizer but also manure application (for both) and management (for CMIP6 only)…"

Comments:

-l618 It is unclear how "the warmer temperature …. allows crops to reach maturity sooner, hence, shortening their grain filling periods" leading to reduced N uptake. The relation between the processes is not clear to me. It is unclear to me the extent to which this hypothesis is supported.

Responses:

After considering the reviewer's comments and reexamining the results, we found that the Asian warmer surface temperature and grain production increases are not uniformly statistically significant and decided that temperature may not be the important factor. We hence removed the concerning discussion from the main text.

Comments:

-l671-672 "These new features enabled CESM2 to perform, for the first time, a more reliable estimation of soil NH3 emission and atmospheric NH3 concentration than using constant emission inventory values under dynamic climate and environmental conditions." Vira et al. (2021) also included the coupling between soil and atmosphere.

Responses:

Vira et al. (2021) was not published when this paper was written and submitted for discussion. We agree to remove the phrase "for the first time" in the sentence.

Comments:

-l706: "We did not include manure application". I thought manure application was indeed included.

Responses:

Thank you. We have corrected the typo:

"We did not include manure **management** in our study due to the high uncertainty and data insufficiency for validation."

-l720 There are global pH datasets that can be used!

Thank you for pointing this out. We are aware of some of the coarse resolution options, but we would still like to advocate for the necessary of field surveys, especially for places with low data coverage. We revised the sentence to emphasize that:

"We also note that such field surveys, **especially in underrepresented regions with low data coverage**, would also be useful to infer a soil pH map that constraints the uncertainty in simulations using a constant pH, like those reported in this study."